# Multimodal brain age estimates relate to Alzheimer disease biomarkers and cognition in early stages: a cross-sectional observational study

Peter R Millar[1]*, Brian A Gordon[2], Patrick H Luckett[3], Tammie LS Benzinger[2,3], Carlos Cruchaga[4], Anne M Fagan[1], Jason J Hassenstab[1], Richard J Perrin[1,5], Suzanne E Schindler[1], Ricardo F Allegri[6], Gregory S Day[7], Martin R Farlow[8], Hiroshi Mori[9], Georg Nübling[10,11], The Dominantly Inherited Alzheimer Network, Randall J Bateman[1], John C Morris[1], Beau M Ances[1,2]

[1]Department of Neurology, Washington University in St. Louis, St Louis, United States; [2]Department of Radiology, Washington University in St. Louis, St Louis, United States; [3]Department of Neurosurgery, Washington University in St. Louis, St Louis, United States; [4]Department of Psychiatry, Washington University in St. Louis, St Louis, United States; [5]Department of Pathology and Immunology, Washington University in St. Louis, St Louis, United States; [6]Department of Cognitive Neurology, Institute for Neurological Research (FLENI), Buenos Aires, Argentina; [7]Department of Neurology, Mayo Clinic Florida, Jacksonville, United States; [8]Department of Neurology, Indiana University School of Medicine, Indianapolis, United States; [9]Department of Clinical Neuroscience, Osaka Metropolitan University Medical School, Nagaoka Sutoku University, Osaka, Japan; [10]Department of Neurology, Ludwig-Maximilians University, Munich, Germany; [11]German Center for Neurodegenerative Diseases, Munich, Germany

*For correspondence:
pmillar@wustl.edu

Group author details:
The Dominantly Inherited
Alzheimer Network See page 18

Competing interest: See page
18

Reviewing Editor: Karla L Miller,
University of Oxford, United
Kingdom

## Abstract

**Background:** Estimates of 'brain-predicted age' quantify apparent brain age compared to normative trajectories of neuroimaging features. The brain age gap (BAG) between predicted and chronological age is elevated in symptomatic Alzheimer disease (AD) but has not been well explored in presymptomatic AD. Prior studies have typically modeled BAG with structural MRI, but more recently other modalities, including functional connectivity (FC) and multimodal MRI, have been explored.

**Methods:** We trained three models to predict age from FC, structural (S), or multimodal MRI (S+FC) in 390 amyloid-negative cognitively normal (CN/A−) participants (18–89 years old). In independent samples of 144 CN/A−, 154 CN/A+, and 154 cognitively impaired (CI; CDR > 0) participants, we tested relationships between BAG and AD biomarkers of amyloid and tau, as well as a global cognitive composite.

**Results:** All models predicted age in the control training set, with the multimodal model outperforming the unimodal models. All three BAG estimates were significantly elevated in CI compared to controls. FC-BAG was significantly reduced in CN/A+ participants compared to CN/A−. In CI participants only, elevated S-BAG and S+FC BAG were associated with more advanced AD pathology and lower cognitive performance.

**Conclusions:** Both FC-BAG and S-BAG are elevated in CI participants. However, FC and structural MRI also capture complementary signals. Specifically, FC-BAG may capture a unique biphasic

response to presymptomatic AD pathology, while S-BAG may capture pathological progression and cognitive decline in the symptomatic stage. A multimodal age-prediction model improves sensitivity to healthy age differences.

**Funding:** This work was supported by the National Institutes of Health (P01-AG026276, P01- AG03991, P30-AG066444, 5-R01-AG052550, 5-R01-AG057680, 1-R01-AG067505, 1S10RR022984-01A1, and U19-AG032438), the BrightFocus Foundation (A2022014F), and the Alzheimer's Association (SG-20-690363-DIAN).

## Editor's evaluation

This is a useful study exploring multi-modality brain age (structural plus resting state MRI) in people in the early stages or at risk of Alzheimer's disease. They found solid evidence that people with cognitive impairment had older-appearing brains and that older-appearing brains were related to Alzheimer's risk factors such as amyloid and tau deposition. Their data suggest that the multi-modality brain age model is more accurate than a unimodal structural MRI model.

## Introduction

Alzheimer disease (AD) is marked by structural and functional disruptions in the brain, some of which can be observed through multimodal magnetic resonance imaging (MRI) in preclinical and symptomatic stages of the disease (*Frisoni et al., 2010*; *Brier et al., 2014a*). More recently, the 'brain-predicted age' framework has emerged as a promising tool for neuroimaging analyses, leveraging recent developments and accessibility of machine-learning techniques, as well as large-scale, publicly available neuroimaging datasets (*Cole and Franke, 2017b*; *Franke and Gaser, 2019*). These models are trained to quantify how 'old' a brain appears, as compared to a normative sample of training data - typically consisting of cognitively normal participants across the adult lifespan (e.g., *Cole et al., 2015*). Thus, the framework allows for a residual-based interpretation of the brain age gap (BAG), defined as the difference between model-predicted age and chronological age, as an index of vulnerability and/or resistance to underlying disease pathology. Indeed, several studies have demonstrated that BAG is elevated (i.e. the brain 'appears older' than expected) in a host of neurological and psychiatric disorders, including symptomatic AD (*Franke et al., 2010*; *Franke and Gaser, 2012*; *Gaser et al., 2013*), as well as schizophrenia (e.g., *Koutsouleris et al., 2014*), HIV (e.g., *Cole et al., 2017c*), and type-2 diabetes (e.g., *Franke et al., 2013*), and moreover, predicts mortality (*Cole et al., 2018*). Conversely, lower BAG is associated with lower risk of disease progression (*Gaser et al., 2013*; *Wang et al., 2019*; *Bocancea et al., 2021*). Critically, at least one comparison suggests that BAG exceeds other established MRI (hippocampal volume) and CSF (pTau and Aβ42) biomarkers in sensitivity to AD progression (*Gaser et al., 2013*). Thus, by summarizing complex, non-linear, highly multivariate patterns of neuroimaging features into a simple, interpretable summary metric, BAG may reflect a comprehensive biomarker of brain health.

Several studies have established that symptomatic AD and mild cognitive impairment (MCI) are associated with elevated BAG (*Cole and Franke, 2017b*; *Franke and Gaser, 2019*). However, the sensitivity of these model estimates to AD in the presymptmatic stage (i.e. present amyloid pathology in the absence of cognitive decline [*Sperling et al., 2011*]) is less clear. The development of sensitive, reliable, non-invasive biomarkers of preclinical AD pathology is critical for the assessment of individual AD risk, as well as the evaluation of AD clinical prevention trials. Recent studies have demonstrated that greater BAG is associated with greater amyloid PET burden in a Down syndrome cohort (*Cole et al., 2017a*) and with greater tau PET burden in sporadic MCI and symptomatic AD (*Lee et al., 2022*). One approach to maximize sensitivity of BAG to presymptomatic AD pathology may be to train brain age models exclusively on amyloid-negative participants. As undetected AD pathology might influence MRI measures, and thus confound effects otherwise attributed to 'healthy aging' (*Brier et al., 2014b*), including the patterns learned by a traditional brain age model, an alternative model trained on amyloid-negative participants only might be more sensitive to detect presymptomatic AD pathology as deviations in BAG. Indeed, one recent study demonstrated that an amyloid-negative trained brain age model (*Ly et al., 2020*) is more sensitive to progressive stages of AD than a typical amyloid-insensitive model (*Cole et al., 2015*).

**eLife digest** The brains of people with advanced Alzheimer's disease often look older than expected based on the patients' actual age. This 'brain age gap' (how old a brain appears compared to the person's chronological age) can be calculated thanks to machine learning algorithms which analyse images of the organ to detect changes related to aging. Traditionally, these models have relied on images of the brain structure, such as the size and thickness of various brain areas; more recent models have started to use activity data, such as how different brain regions work together to form functional networks.

While the brain age gap is a useful measure for researchers who investigate aging and disease, it is not yet helpful for clinicians. For example, it is unclear whether the machine learning algorithm could detect changes in the brains of individuals in the initial stages of Alzheimer's disease, before they start to manifest cognitive symptoms.

Millar et al. explored this question by testing whether models which incorporate structural and activity data could be more sensitive to these early changes. Three machine learning algorithms (relying on either structural data, activity data, or combination of both) were used to predict the brain ages of participants with no sign of disease; with biological markers of Alzheimer's disease but preserved cognitive functions; and with marked cognitive symptoms of the condition.

Overall, the combined model was slightly better at predicting the brain age of healthy volunteers, and all three models indicated that patients with dementia had a brain which looked older than normal. For this group, the model based on structural data was also able to make predictions which reflected the severity of cognitive decline. Crucially, the algorithm which used activity data predicted that, in individuals with biological markers of Alzheimer's disease but no cognitive impairment, the brain looked in fact younger than chronological age. Exactly why this is the case remains unclear, but this signal may be driven by neural processes which unfold in the early stages of the disease. While more research is needed, the work by Millar et al. helps to explore how various types of machine learning models could one day be used to assess and predict brain health.

However, this comparison included amyloid-negative and amyloid-positive test samples from two separate cohorts and thus may be driven by cohort, scanner, and/or site differences. To validate the applicability of the brain-predicted age approach to presymptomatic AD, it is important to test a model's sensitivity to amyloid status, as well as continuous relationships with AD biomarkers, within a single cohort. Another recent comparison demonstrated that both traditional and amyloid-negative trained brain age models were similarly related to molecular AD biomarkers, but that further attempts to 'disentangle' AD from brain age by including more advanced AD continuum participants in the training sample significantly reduced relationships between brain age and AD markers (*Hwang et al., 2022*). Thus, in this study, we will apply the amyloid-negative training approach to a multimodal MRI dataset in order to maximize sensitivity to AD pathology in the presymptomatic stage.

Most of the brain-predicted age reports described above focused primarily on structural MRI. However, other studies have successfully modeled brain age using a variety of other modalities, including metabolic PET (*Goyal et al., 2019*; *Lee et al., 2022*), diffusion MRI (*Cherubini et al., 2016*; *Petersen et al., 2022*), and functional connectivity (FC) (*Dosenbach et al., 2010*; *Liem et al., 2017*; *Eavani et al., 2018*; *Nielsen et al., 2019*). Integration of multiple neuroimaging modalities may maximize sensitivity of BAG estimates to preclinical AD. Indeed, recent multimodal comparisons suggest that structural MRI and FC capture complementary age-related signals (*Eavani et al., 2018*; *Dunås et al., 2021*) and that age prediction may be improved by incorporating multiple modalities (*Liem et al., 2017*; *Engemann et al., 2020*). One recent study has shown that BAG estimates from an FC graph theory-based model are significantly elevated in autosomal dominant AD mutation carriers and are positively associated with amyloid PET (*Gonneaud et al., 2021*). Furthermore, we have recently demonstrated that FC correlation-based BAG estimates are surprisingly reduced in cognitively normal participants with evidence of amyloid pathology and elevated pTau, as well as in cognitively normal APOE ε4 carriers at genetic risk of AD (*Millar et al., 2022*). Thus, incorporating FC into BAG models may improve sensitivity to early AD.

This project aimed to develop multimodal models of brain-predicted age, incorporating both FC and structural MRI. Participants with presymptomatic AD pathology were excluded from the training set to maximize sensitivity. We hypothesized that BAG estimates would be sensitive to the presence of AD biomarkers and early cognitive impairment. We further considered whether estimates were continuously associated with AD biomarkers of amyloid and tau, as well as cognition. We hypothesized that FC and structural MRI would capture complementary signals related to age and AD. Thus, we systematically compared models trained on unimodal FC, structural MRI, and combined modalities to test the added utility of multimodal integration in accurately predicting age and whether each modality captures unique relationships with AD biomarkers and cognition.

## Methods

### Participants

We formed a training sample of healthy controls spanning the adult lifespan by combining structural and FC-MRI data from three sources, as described previously (*Millar et al., 2022*): the Charles F. and Joanne Knight AD Research Center (ADRC) at Washington University in St. Louis (WUSTL), healthy controls from studies in the Ances lab at WUSTL (*Thomas et al., 2013*; *Petersen et al., 2021*), and mutation-negative controls from the Dominantly Inherited Alzheimer Network (DIAN) study of autosomal dominant AD at multiple international sites including WUSTL (*McKay et al., 2022*). To minimize the likelihood of undetected AD pathology in our training set, participants over the age of 50 were only included in the training set if they were cognitively normal, as assessed by the Clinical Dementia Rating (CDR 0; *Morris, 1993*), and had at least one biomarker indicating the absence of amyloid pathology (CN/A−, see below). We excluded 59 participants who did not have available CDR or biomarker measures (see *Figure 1—figure supplement 1*). As CDR and amyloid biomarkers were not available in the Ances lab controls, we included only participants at or below age 50 from this cohort in the training set. These healthy control participants were randomly divided into a training set (~80%; N=390) and a held-out test set (~20%; N=97), which did not significantly differ in age, sex, education, or race, see *Table 1*.

Finally, independent samples for hypothesis testing included three groups from the Knight ADRC: a randomly selected sample of 144 CN/A− controls who did not overlap with the training or testing sets, 154 CN/A+ participants, and 154 cognitively impaired (CI) participants (CDR > 0 with a biomarker measure consistent with amyloid pathology [see below] and/or a primary diagnosis of AD or uncertain dementia [*McKhann et al., 2011*]). See *Table 1* for demographic details of each sample. All participants provided written informed consent in accordance with the Declaration of Helsinki and their local institutional review board. All procedures were approved by the Human Research Protection Office at WUSTL (IRB ID # 201204041).

### PET and CSF biomarkers

Amyloid burden was imaged with PET using (11 C)-Pittsburgh Compound B (PIB; *Klunk et al., 2004*) or (18 F)-Florbetapir (AV45; *Wong et al., 2010*). Regional standard uptake ratios (SUVRs) were modeled from 30 to 60 min after injection for PIB and from 50 to 70 min for AV45, using cerebellar gray as the reference region (*Su et al., 2013*). Regions of interest were segmented automatically using FreeSurfer 5.3 (*Fischl, 2012*). Global amyloid burden was defined as the mean of partial-volume-corrected (PVC) SUVRs from bilateral precuneus, superior and rostral middle frontal, lateral and medial orbitofrontal, and superior and middle temporal regions (*Su et al., 2013*). Amyloid summary SUVRs were harmonized across tracers using a centiloid conversion (*Su et al., 2018*).

Tau deposition was imaged with PET using (18 F)-Flortaucipir (AV-1451; *Chien et al., 2013*). Regional SUVRs were modeled from 80 to 100 min after injection, using cerebellar gray as the reference region. A tau summary measure was defined in the mean PVC SUVRs from bilateral amygdala, entorhinal, inferior temporal, and lateral occipital regions (*Mishra et al., 2017*).

CSF was collected via lumbar puncture using methods described previously (*Fagan et al., 2006*). After overnight fasting, 20–30 mL samples of CSF were collected, centrifuged, then aliquoted (500 µL) in polypropylene tubes, and stored at –80°C. CSF amyloid β peptide 42 (Aβ42), Aβ40, and phosphorylated tau-181 (pTau) were measured with automated Lumipulse immunoassays (Fujirebio, Malvern, PA, USA) using a single lot of assays for each analyte. Aβ42 and pTau estimates were each normalized

**Table 1.** Demographic information of the combined samples.

| Measure | Training sets (total N=390) | | | Test sets (total N=97)§ | | | Analysis sets (total N=452) | | |
|---|---|---|---|---|---|---|---|---|---|
| | Ances Controls (CN/<50) | DIAN Controls (CN/A−) | Knight ADRC Controls (CN/A−) | Ances Controls (CN/<50) | DIAN Controls (CN/A−) | Knight ADRC Controls (CN/A−) | CN/A− | CN/A+ | CI |
| N | 136 | 120 | 134 | 38 | 26 | 33 | 144 | 154 | 154 |
| Age (mean, SD) | 29.92 (9.92) | 40.02 (10.26) | 64.97 (10.57) | 26.68 (7.11) | 41.46 (12.34) | 64.73 (10.57) | 66.93 (8.53) | 72.56 (7.15)‡ | 75.67 (6.86) ‡ |
| CDR (N 0 / N 0.5 / N 1.0 / N 2.0) | NA | 120 / 0 / 0 / 0 | 134 / 0 / 0 / 0 | NA | 26 / 0 / 0 / 0 | 33 / 0 / 0 / 0 | 144 / 0 / 0 / 0 | 154 / 0 / 0 / 0 | 0 / 119 / 35 / 2 |
| Amyloid status (N + / N -) | NA | 120 / 0 | 134 / 0 | NA | 26 / 0 | 33 / 0 | 144 / 0 | 0 / 154 | 0 / 57 |
| Biomarkers available (N PET / CSF / both) | NA | 30 / 6 / 79 | 11 / 22 / 91 | NA | 3 / 1 / 21 | 5 / 0 / 28 | 24 / 0 / 120 | 17 / 0 / 137 | 14 / 0 / 43 |
| APOE ε4 carrier status (N + / N -) | NA | 76 / 44 | 99 / 34 | NA | 19 / 7 | 28 / 5 | 115 / 29 | 71 / 83 ‡ | 55 / 98 ‡ |
| MMSE (mean, SD) | NA | NA | 29.26 (1.05) | NA | NA | 29.45 (0.94) | 29.13 (1.17) | 28.97 (1.33) | 25.37 (3.55) ‡ |
| Sex (N female / N male) | 70 / 64 | 85 / 35 | 84 / 50 | 19 / 18 | 16 / 10 | 22 / 11 | 89 / 55 | 91 / 63 | 68 / 86† |
| **Years of education (mean, SD)** | 13.68 (2.16) | 14.78 (3.04) | 16.16 (2.43) | 13.95 (1.99) | 14.92 (2.83) | 16.48 (2.43) | 15.71 (2.65) | 15.90 (2.64) | 15.05 (2.97)* |
| Race (N American Indian or Alaska Native) | 1 | 0 | 0 | 1 | 0 | 0 | 0 | 0 | 0 |
| Race (N Asian) | 1 | 1 | 2 | 0 | 0 | 0 | 0 | 1 | 0 |
| Race (N Black) | 67 | 0 | 20 | 17 | 0 | 7 | 17 | 16 | 20 |
| Race (N Native Hawaiian or Other Pacific Islander) | 2 | 0 | 0 | 2 | 0 | 0 | 0 | 0 | 0 |
| Race (N White) | 57 | 118 | 112 | 17 | 26 | 26 | 127 | 137 | 134 |
| Site | WUSTL | Multiple sites | WUSTL | WUSTL | Multiple sites | WUSTL | WUSTL | WUSTL | WUSTL |
| Scanner | Siemens Trio | Siemens Trio / Verio | Siemens Trio / Biograph | Siemens Trio | Siemens Trio / Verio | Siemens Trio / Biograph | Siemens Trio / Biograph | Siemens Trio / Biograph | Siemens Trio / Biograph |
| Field strength | 3T | 3T | 3T | 3T | 3T | 3T | 3T | 3T | 3T |

CN = Cognitively Normal, <50 = less than age 50, A− = amyloid negative, A+ = amyloid positive, CI = cognitively Impaired, DIAN = Dominantly Inherited Alzheimer Network, ADRC = Alzheimer Disease Research Center, AD = Alzheimer disease, CDR = Clinical Dementia Rating, MMSE = Mini Mental State Examination, WUSTL = Washington University in St. Louis, T = Tesla. Group differences from the CN/A− analysis set were tested with t tests for continuous variables and $\chi^2$ tests for categorical variables.

*$p < 0.05$, ∧ $p < 0.10$.

†$p < 0.01$.

‡$p < 0.001$.

§Test sets include randomly-selected, non-overlapping subsets of participants drawn from the same studies as the training sets.

for individual differences in CSF production rates by forming a ratio with Aβ40 as the denominator (*Hansson et al., 2019*; *Guo et al., 2020*). As pTau/Aβ40 was highly skewed, we applied a log transformation to these estimates before statistical analysis.

Amyloid positivity was defined using previously published cutoffs for PIB (SUVR > 1.42; *Vlassenko et al., 2016*) or AV45 (SUVR > 1.19; *Su et al., 2019*). Additionally, the CSF Aβ42/Aβ40 ratio has been shown to be highly concordant with amyloid PET (positivity cutoff < 0.0673; *Schindler et al., 2018*; *Volluz et al., 2021*). Thus, participants were defined as amyloid-positive (for CN/A+ and CI groups) if they had either a PIB, AV45, or CSF Aβ42/Aβ40 ratio measure in the positive range. Participants with discordant positivity between PET and CSF estimates were defined as amyloid-positive.

## Cognitive battery

Knight ADRC participants completed a 2 hr battery of cognitive tests. We examined global cognition by forming a composite of tasks across cognitive domains, including processing speed (Trail Making A; *Schindler et al., 2018*), executive function (Trail Making B; *Schindler et al., 2018*), semantic fluency (Animal Naming; *Armitage, 1946*), and episodic memory (Free and Cued Selective Reminding Test free recall score; *Goodglass and Kaplan, 1983*; *Grober et al., 1988*). This composite has recently been used to study individual differences in cognition in relation the preclinical AD biomarkers and structural MRI (*Aschenbrenner et al., 2018*), as well as functional MRI measures (*Millar et al., 2021*).

## MRI acquisition

All MRI data were obtained using a Siemens 3T scanner, although there was a variety of specific models within and across studies. As described previously (*Millar et al., 2022*), participants in the Knight ADRC and Ances lab studies completed one of two comparable structural MRI protocols, varying by scanner (sagittal T1-weighted magnetization-prepared rapid gradient echo sequence [MPRAGE] with repetition time [TR] = 2400 or 2300 ms, echo time [TE] = 3.16 or 2.95 ms, flip angle = 8 or 9°, frames = 176, field of view = sagittal 256×256 or 240×256 mm, 1 mm isotropic or 1×1×1.2 mm voxels; oblique T2-weighted fast spin echo sequence [FSE] with TR = 3200 ms, TE = 455 ms, 256×256 acquisition matrix, 1 mm isotropic voxels) and an identical resting-state fMRI protocol (interleaved whole-brain echo planar imaging sequence [EPI] with TR = 2200 ms, TE = 27 ms, flip angle = 90°, field of view = 256 mm, 4 mm isotropic voxels for two 6 min runs [164 volumes each] of eyes open fixation). DIAN participants completed a similar MPRAGE protocol (TR = 2300ms, TE = 2.95ms, flip angle = 9°, field of view = 270 mm, 1.1×1.1×1.2 mm voxels; *McKay et al., 2022*). Resting-state EPI sequence parameters for the DIAN participants differed across sites and scanners with the most notable difference being shorter resting-state runs (one 5 min run of 120 volumes; see *Supplementary file 1* for summary of structural and functional MRI parameters; *McKay et al., 2022*).

## FC preprocessing and features

All MRI data were processed using common pipelines. Initial fMRI preprocessing followed conventional methods, as described previously (*Shulman et al., 2010*; *Millar et al., 2022*), including frame alignment, debanding, rigid body transformation, bias field correction, and normalization of within-run intensity values to a whole-brain mode of 1000 (*Power et al., 2012*). Transformation to an age-appropriate in-house atlas template (based on independent samples of either younger adults or CN older adults) was performed using a composition of affine transforms connecting the functional volumes with the T2-weighted and MPRAGE images. Frame alignment was included in a single resampling that generated a volumetric time series of the concatenated runs in isotropic 3 mm atlas space.

As described previously (*Fox et al., 2009*; *Millar et al., 2022*), additional processing was performed to allow for nuisance variable regression. Data underwent framewise censoring based on motion estimates (framewise displacement [FD] > 0.3 mm and/or derivative of variance [DVARS] > 2.5 above participant's mean). To further minimize the confounding influence of head motion on FC estimates (*Power et al., 2012*) in all samples, we only included scans with low head motion (mean FD < 0.30 mm and > 50% frames retained after motion censoring). BOLD data underwent a temporal band-pass filter (0.005 Hz < $f$ < 0.1 Hz) and nuisance variable regression, including motion parameters, timeseries from FreeSurfer 5.3-defined (*Fischl, 2012*) whole brain (global signal), CSF, ventricle, and white matter masks, as well as the derivatives of these signals. Finally, BOLD data were spatially blurred (6 mm full width at half maximum).

Final BOLD time series data were averaged across voxels within a set of 300 spherical regions of interest (ROIs) in cortical, subcortical, and cerebellar areas (*Seitzman et al., 2020*). For each scan, we calculated the 300×300 Fisher-transformed Pearson correlation matrix of the final averaged BOLD time series between all ROIs. We then used the vectorized upper triangle of each correlation matrix (excluding auto-correlations; 44,850 total correlations) as input features for predicting age. Since site and/or scanner differences between samples might confound neuroimaging estimates, we harmonized FC matrices using an empirical Bayes modeling approach (ComBat; *Johnson et al., 2007*; *Fortin et al., 2017*), which has previously been applied to FC data (*Yu et al., 2018*).

## Structural MRI processing and features

All T1-weighted images underwent cortical reconstruction and structural segmentation through a common pipeline with FreeSurfer 5.3 (*Fischl et al., 2002*; *Fischl, 2012*). Structural processing included segmentation of subcortical white matter and deep gray matter, intensity normalization, registration to a spherical atlas, and parcellation of the cerebral cortex based on the Desikan atlas (*Desikan et al., 2006*). Inclusion and exclusion errors of parcellation and segmentation were identified and edited by a centralized team of trained research technicians according to standardized criteria (*Su et al., 2013*). We then used the FreeSurfer-defined thickness estimates from 68 cortical regions (*Desikan et al., 2006*), along with volume estimates from 33 subcortical regions (*Fischl et al., 2002*) as input features for predicting age. We harmonized structural features across sites and scanners using the same ComBat approach (*Johnson et al., 2007*; *Fortin et al., 2017*), which has also been applied to structural MRI data (*Fortin et al., 2018*).

## Gaussian process regression

As described previously (*Millar et al., 2022*), machine-learning analyses were conducted using the Regression Learner application in Matlab (*MathWorks, 2021*). We trained two Gaussian process regression (GPR; *Rasmussen et al., 2004*) models, each with a rational quadratic kernel function to predict chronological age using fully-processed, harmonized MRI features (FC or structural) in the training set. The $\sigma$ hyperparameter was tuned within each model by searching a range of values from $10^{-4}$ to $10*SD_{age}$ using Bayesian optimization across 100 training evaluations. The optimal value of $\sigma$ for each model was found (see *Figure 1—figure supplement 2*) and was applied for all subsequent applications of that model. All other hyperparameters were set to default values (basis function = constant and standardize = true).

Model performance in the training set was assessed using 10-fold cross validation via the Pearson correlation coefficient ($r$), the proportion of variance explained ($R^2$), the mean absolute error ($MAE$), and root-mean-square error ($RMSE$) between true chronological age and the cross-validated age predictions merged across the 10 folds. We then evaluated generalizability of the models to predict age in unseen data by applying the trained models to the held-out test set of healthy controls. Finally, we applied the same fully-trained GPR models to separate analysis sets of 154 CI, 154 CN/A+, and 144 CN/A− controls to test our hypotheses regarding AD-related group effects and individual difference relationships. Unimodal models were each constructed with a single GPR model. The multimodal model was constructed by taking the 'stacked' predictions from each first-level unimodal model as features for training a second-level GPR model (*Liem et al., 2017*; *Engemann et al., 2020*; *Dunås et al., 2021*).

For each participant, we calculated model-specific BAG estimates as the difference between chronological age and age predictions from the unimodal FC model (FC-BAG), structural model (S-BAG), and multimodal model (S+FC BAG). To correct for regression dilution commonly observed in similar models (*Le et al., 2018*; *Smith et al., 2019*; *Liang et al., 2019*), we included chronological age as a covariate in all statistical tests of BAG (*Cole et al., 2017a*; *Le et al., 2018*). However, to avoid inflating estimates of prediction accuracy (*Butler et al., 2021*), only *uncorrected* age prediction values were used for evaluating model performance in the training and test sets.

## Statistical analysis

All statistical analyses were conducted in R 4.0.2 (*R Development Core Team, 2020*). Demographic differences in the AD samples were tested with independent-samples $t$ tests for continuous variables and $\chi^2$ tests for categorical variables, using CN/A− controls as a reference group. Differences

in brain age model performance were tested using Williams's test of difference between dependent correlations sharing one variable, i.e., Pearson's $r$ between age and each model prediction of age. To correct for age-related bias in BAG (*Le et al., 2018*; as previously mentioned), we controlled for age as a covariate during all statistical tests. Group differences in each BAG estimate were tested using an omnibus ANOVA test with follow-up pairwise $t$ tests on age-residualized BAG estimates, using a false discovery rate (FDR) correction for multiple comparisons. Assumptions of normality were tested by visual inspection of quantile-quantile plots. Assumptions of equality of variance were tested with Levene's test. Linear regression models tested the effects of cognitive impairment (CDR > 0 vs. CDR 0) and amyloid positivity (A− vs. A+) on BAG estimates from each model, controlling for true age (as noted above), sex, and years of education. Given the potential confounding influence of head motion on FC-derived measures (*Power et al., 2012*; *Van Dijk et al., 2012*; *Satterthwaite et al., 2012*), we also included mean FD as an additional covariate of non-interest in the FC and S+FC models. We tested continuous relationships with AD biomarkers and cognitive estimates using linear regression models, including the same demographic and motion covariates. Since the range of amyloid biomarkers was drastically reduced in the CN/A− sample, we excluded these participants from models testing continuous amyloid relationships. Effect sizes were computed as partial $\eta^2$ ($\eta_p^2$).

## Results

### Sample description and demographics

Demographic characteristics of the training sets, test sets, and analysis sets are reported in *Table 1*. CN/A+ participants were older ($t = 6.15$, $p < 0.001$) and more likely to be *APOE* ε4 carriers ($\chi^2 = 34.73$, $p < 0.001$) than amyloid-negative controls. Furthermore, CI participants were older ($t = 9.71$, $p < 0.001$), more likely male ($\chi^2 = 8.60$, $p = 0.003$), more likely to be *APOE* ε4 carriers ($\chi^2 = 56.67$, $p < 0.001$), and had fewer years of education ($t = 2.03$, $p < 0.043$), and lower MMSE scores ($t = 12.46$, $p < 0.001$) than amyloid-negative controls.

### Comparison of model performance

All models accurately predicted chronological age in the training sets, as assessed using 10-fold cross validation, as well as in the held-out test sets. Overall, prediction accuracy was lowest in the FC model ($MAE_{FC/Train} = 8.67$ years, $R^2_{FC/Train} = 0.68$, $MAE_{FC/Test} = 8.25$ years, $R^2_{FC/Test} = 0.73$; see *Figure 1A & B*). The structural MRI model ($MAE_{S/Train} = 5.97$ years, $R^2_{S/Train} = 0.81$, $MAE_{S/Test} = 6.26$ years, $R^2_{S/Test} = 0.82$; see *Figure 1C & D*) significantly outperformed the FC model in age prediction accuracy, Williams's $t_{S\ vs.\ FC} = 5.39$, $p < 0.001$. There was a significant, but modestly sized, positive correlation between FC-BAG and S-BAG in the adult lifespan CN/A− training and testing sets ($r = 0.095$, $p = 0.036$; see *Figure 1— figure supplement 3A*), as well as the AD analysis sets ($r = 0.134$, $p = 0.004$; see *Figure 1—figure supplement 3B*).

Finally, the multimodal model ($MAE_{S+FC/Train} = 5.34$ years, $R^2_{S+FC/Train} = 0.86$, $MAE_{S+FC/Test} = 5.25$ years, $R^2_{S+FC/Test} = 0.87$; see *Figure 1E & F*) significantly outperformed both the FC model (Williams's $t_{S+FC\ vs.\ FC} = 11.20$, $p < 0.001$) and the structural MRI model (Williams's $t_{S+FC\ vs.\ S} = 5.67$, $p < 0.001$). It is possible that the modest increase in the multimodal model was due to capitalizing on noise, simply by adding more features to the structural model. Hence, we also compared the observed $R^2_{S+FC}$ to a bootstrapped distribution of $R^2$ performance estimates from 1000 resamples using a model in which the original structural MRI model was stacked with a model trained on randomly reshuffled FC features. Thus, this distribution represents the expected improvements in model performance from simply adding new features to the structural MRI model with the stacked approach. The observed $R^2_{S+FC}$ outperformed all $R^2$ estimates from this bootstrapped distribution ($p < 0.001$; see *Figure 1—figure supplement 4*), suggesting that the modest increase in model performance observed in the stacked multimodal (S+FC) model over the unimodal structural model is due to meaningful age-related FC signal, rather than capitalizing on noise in a larger feature set.

### BAG differences in cognitive impairment and amyloid positivity

Residual FC-BAG was normally distributed (see *Figure 2—figure supplement 1*), and variance in FC-BAG did not significantly differ between the analysis sets, Levene's statistic = 0.01, $p = 0.988$. An omnibus ANOVA revealed significant differences in residual FC-BAG across the three groups,

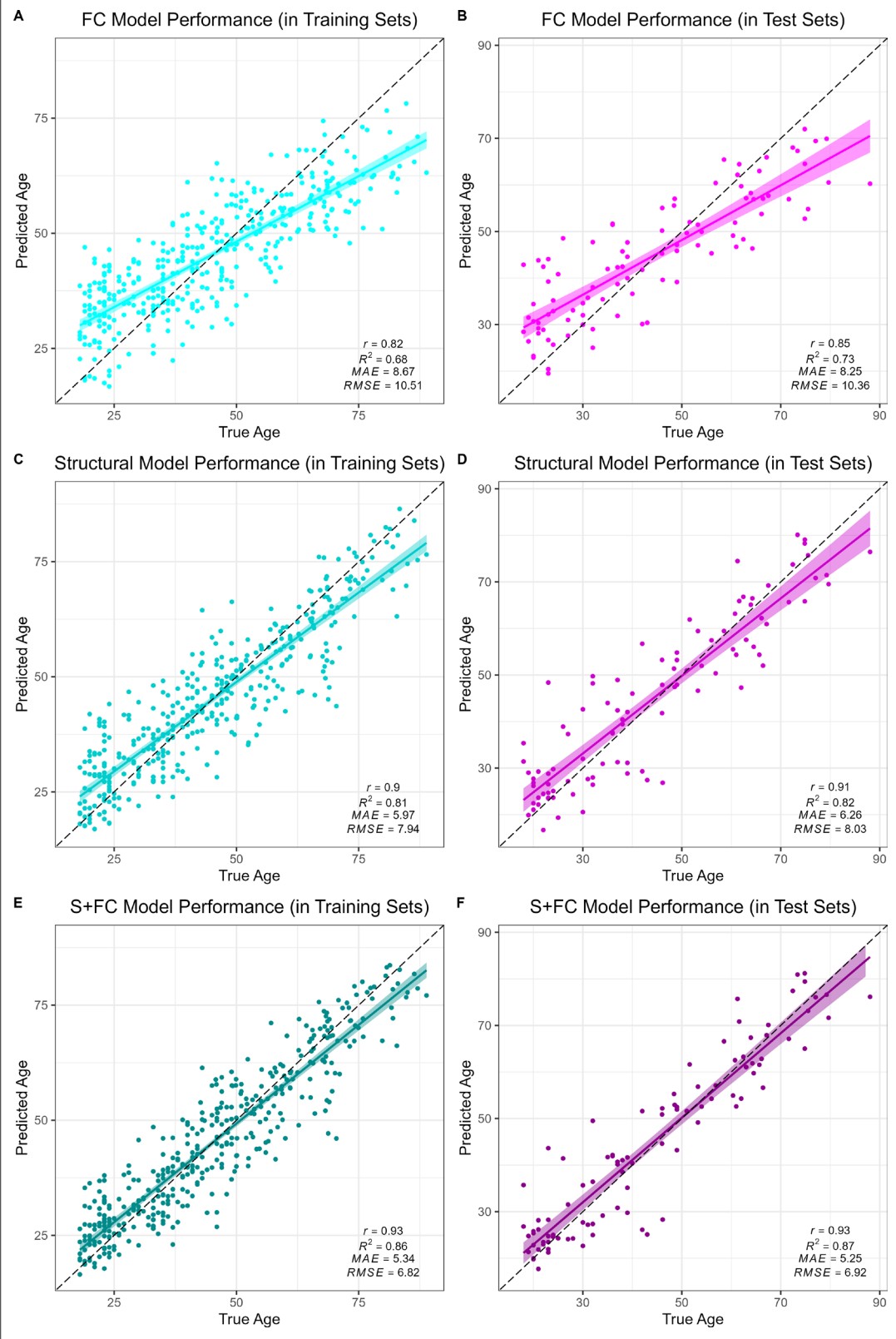

**Figure 1.** Performance of the brain age models in the training (left column) and test sets (right column) for each modality: functional connectivity (FC; **A and B**), structural MRI (S; **C and D**) and multimodal models (S+FC; **E and F**). Age predicted by each model (y axis) is plotted against true age (x axis). Colored lines and shaded areas represent regression lines and 95% confidence regions. Dashed black lines represent perfect prediction. Model

*Figure 1 continued on next page*

*Figure 1 continued*

performance is evaluated by Pearson's *r*, proportion of variance explained ($R^2$), mean absolute error (*MAE*), and root-mean-square error (*RMSE*).

The online version of this article includes the following figure supplement(s) for figure 1:

**Figure supplement 1.** Flow chart of participant inclusion, exclusion, and group assignments.

**Figure supplement 2.** Tuning curves of $\sigma$ hyperparameter in training for structural (**A**) and functional connectivity (**B**) Gaussian process regression (GPR) models.

**Figure supplement 3.** Correlation between S-brain age gap (BAG; x axis) and functional connectivity (FC)-BAG (y axis) estimates in the training and validation sets (**A**) and analysis sets (**B**).

**Figure supplement 4.** Violin plot of $R^2$ performance estimates from 1000 bootstrapped samples in which a stacked brain age model combined the fully-trained structural MRI model ($R^2_S$) with a reshuffled functional connectivity (FC) model (i.e. FC training features were randomly reassigned in each bootstrap sample).

$F(2,449)$ = 9.80, $p$ < 0.001. FC-BAG was 2.17 years older in CI participants compared to CN controls ($\beta$ = 2.17, $p$ = 0.030, $\eta_p^2$ = 0.01; see *Figure 2A&B*, *Table 2A*). Follow-up *t* tests revealed that residual FC-BAG was significantly elevated in CI relative to CN/A+participants ($p_{FDR}$ < 0.001). FC-BAG was also 1.64 years lower in A+ participants compared to A− ($\beta$ = −1.64, $p$ = 0.035, $\eta_p^2$ = 0.01), controlling for global CDR and the other covariates. Follow-up *t* tests revealed that residual FC-BAG was significantly lower in CN/A+ participants compared to CN/A− controls ($p_{FDR}$ = 0.002).

Residual S-BAG was also normally distributed (see *Figure 2—figure supplement 1*), and variance in S-BAG did not significantly differ between the analysis sets, Levene's statistic = 0.10, $p$ = 0.902. An omnibus ANOVA revealed significant differences in residual S-BAG across the three groups, $F(2,449)$ = 20.64, $p$ < 0.001. S-BAG was 5.10 years older in CI participants compared to CN controls ($\beta$ = 5.10, $p$ < 0.001, $\eta_p^2$ = 0.04; see *Figure 2C&D*, *Table 2B*). Follow-up *t* tests revealed that residual S-BAG was significantly elevated in CI participants relative to CN/A− and CN/A+ participants ($p_{FDR}$'s < 0.001). S-BAG did not significantly differ as a function of amyloid positivity, controlling for CDR and the other covariates.

Residual S+FC-BAG was also normally distributed (see *Figure 2—figure supplement 1*), and variance in S+FC-BAG did not significantly differ between the analysis sets, Levene's statistic = 0.89, $p$ = 0.412. An omnibus ANOVA revealed significant differences in residual S+FC-BAG across the three groups, $F(2,449)$ = 21.84, $p$ < 0.001. S+FC-BAG was 4.31 years older in CI participants compared to CN controls ($\beta$ = 4.1, $p$ < 0.001, $\eta_p^2$ = 0.04; see *Figure 2E, F*, *Table 2C*). Follow-up *t* tests revealed that residual FC-BAG was significantly elevated in CI participants relative to CN/A− and CN/A+ participants ($p_{FDR}$'s < 0.001). S+FC-BAG did not significantly differ as a function of amyloid positivity, controlling for CDR and the other covariates.

## Relationships with amyloid markers

355 participants (144 CN/A−, 154 CN/A+, 57 CI) had an available amyloid PET scan, and 300 (120 CN/A−, 137 CN/A+, 43 CI) had an available CSF estimate of Aβ42/40. In the FC model, FC-BAG was not significantly related with amyloid PET nor was there an interactive relationship with amyloid PET between groups (see *Figure 3A*). There were also no significant main effects or interactions between FC-BAG, S-BAG, or S+FC BAG and CSF Aβ42/40 (See *Figure 3B, D and F*).

In the structural and multimodal models, there were significant main effects, such that greater S-BAG ($\beta$ = 0.79, $p$ = 0.004, $\eta_p^2$ = 0.041; see *Figure 3C*) and greater S+FC BAG ($\beta$ = 0.81, $p$ = 0.015, $\eta_p^2$ = 0.029; see *Figure 3E*) were both associated with greater amyloid PET. In the multimodal model only, this relationship was further characterized by a non-significant interaction ($\beta$ = 1.16, $p$ = 0.087, $\eta_p^2$ = 0.014), such that the association was significantly positive in CI participants interaction ($\beta$ = 1.53, $p$ = 0.029, $\eta_p^2$ = 0.092) but not in CN/A+ ($\beta$ = −0.05, $p$ = 0.881, $\eta_p^2$ = 0.001).

## Relationships with tau markers

99 participants (42 CN/A−, 40 CN/A+, 17 CI) had an available tau PET scan, and 300 (120 CN/A−, 137 CN/A+, 43 CI) had an available CSF estimate of pTau-181/Aβ40. In the FC model, FC-BAG was not significantly related with tau PET or CSF pTau-181/Aβ40 (see *Figure 4A and B*). However, there was a

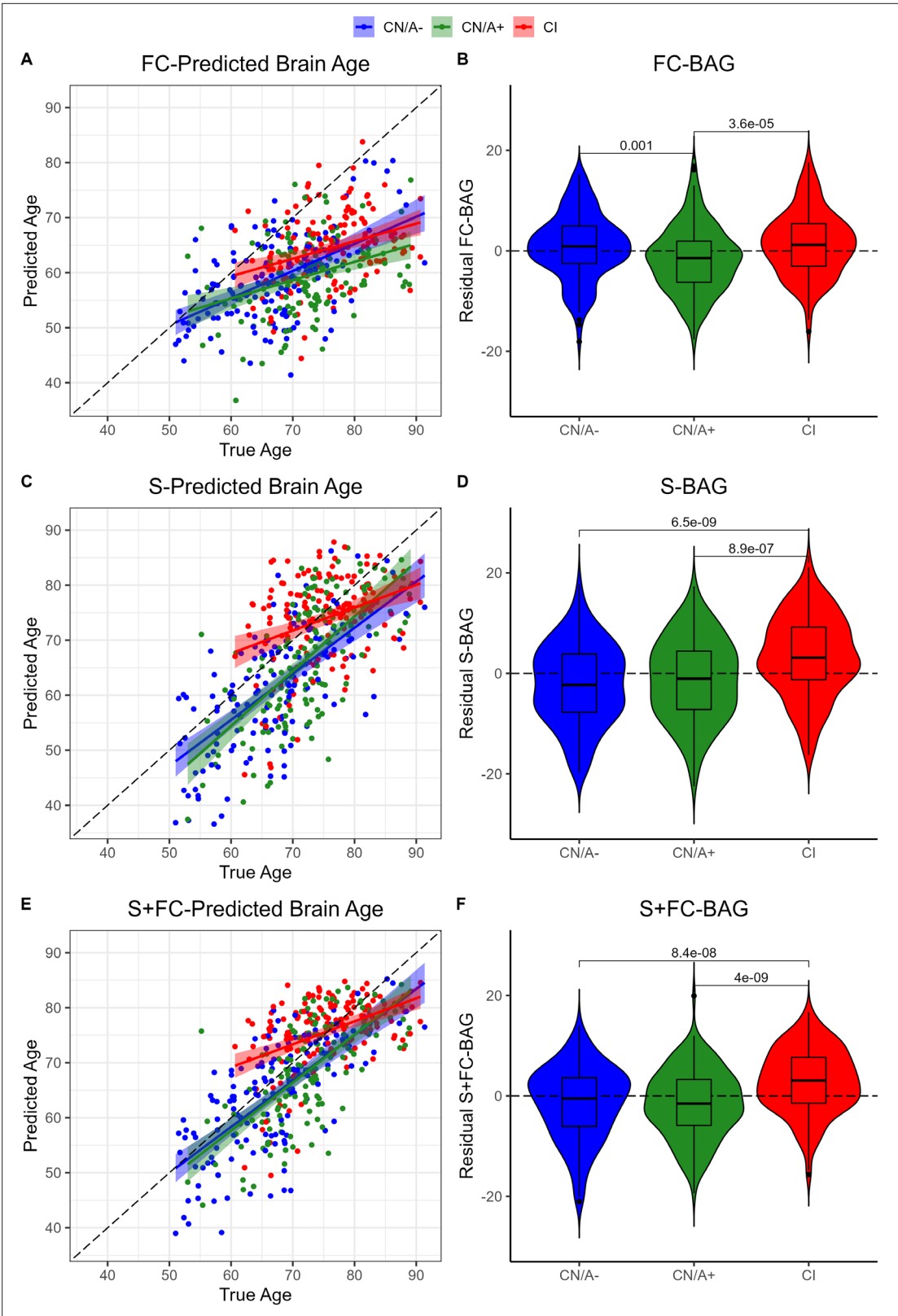

**Figure 2.** Group differences in functional connectivity (FC; **A and B**), structural (S; **C and D**), and multimodal (S+FC; **E and F**) brain age in the analysis sets. Comparisons are presented between cognitively normal (Clinical Dementia Rating [CDR] = 0) biomarker-negative controls (CN/A−; blue) vs. CN/A+ (green) vs. cognitively impaired participants (CI, red). Scatterplots (**A, C, and E**) show predicted vs. true age for each group. Colored lines and

*Figure 2 continued on next page*

*Figure 2 continued*

shaded areas represent group-specific regression lines and 95% confidence regions. Dashed black lines represent perfect prediction. Violin plots (**B, D, and F**) show residual FC-brain age gap (BAG; controlling for true age) in each group. *p* values are reported from pairwise independent-samples *t* tests.

The online version of this article includes the following figure supplement(s) for figure 2:

**Figure supplement 1.** Quantile-quantile plots of brain age gap, controlling for age, in each of the analysis sets (cognitively normal, amyloid negative [CN/A−]; CN/A+; and cognitively impaired [CI]) for functional connectivity [FC; **A**], structural [S; **B**] and multimodal [S+FC; **C**] models.

non-significant interaction, suggesting a more positive association between CSF pTau-181/Aβ40 and FC-BAG in CI participants but not in CN controls ($\beta$ = 0.02, $p$ = 0.059, $\eta_p^2$ = 0.016).

In the structural and multimodal models, there were significant main effects, such that greater S-BAG ($\beta$ = 0.02, $p$ < 0.001, $\eta_p^2$ = 0.141; see ***Figure 4C***) and greater S+FC BAG ($\beta$ = 0.02, $p$ = 0.001, $\eta_p^2$ = 0.110; see ***Figure 4E***) were both associated with greater tau PET. These main effects were further characterized by significant interactions (S-BAG: $\beta$ = 0.04, $p$ < 0.001, $\eta_p^2$ = 0.176; S+FC-BAG: $\beta$ = 0.07, $p$ < 0.001, $\eta_p^2$ = 0.250), such that the positive association was only observed in CI participants, but not in the other groups.

Consistent with tau PET, CSF pTau/Aβ40 demonstrated similar interactive relationships, such that greater S-BAG ($\beta$ = 0.02, $p$ < 0.001, $\eta_p^2$ = 0.052; see ***Figure 4D***) and greater S+FC BAG ($\beta$ = 0.04, $p$ < 0.001, $\eta_p^2$ = 0.075; see ***Figure 4F***) were both associated with greater CSF pTau/Aβ40 in the CI participants, but not in the other groups.

## Relationships with cognition

445 participants (144 CN/A−, 153 CN/A+, 148 CI) had available performance measures from the cognitive composite tasks. In the FC model, there was a significant main effect, such that across all groups, greater FC-BAG was associated with lower cognitive composite score ($\beta$ = −0.01, $p$ = 0.006, $\eta_p^2$ = 0.017; see ***Figure 5A***). However, this effect was driven by group differences in both variables, as there were neither relationships between FC-BAG and cognition within any of the groups nor were there any significant interactions.

In the structural model and multimodal models, there were significant main effects, such that greater S-BAG ($\beta$ = −0.03, $p$ < 0.001, $\eta_p^2$ = 0.104; see ***Figure 5B***) and greater S+FC BAG ($\beta$ = −0.03, $p$ < 0.001, $\eta_p^2$ = 0.096; see ***Figure 5C***) were both associated with lower cognitive composite scores. Both effects were further characterized by significant interactions such that the negative associations were observed in the CI participants, but not in the other groups (S-BAG: $\beta$ = −0.03, $p$ < 0.001, $\eta_p^2$ = 0.045; S+FC-BAG: $\beta$ = −0.04, $p$ < 0.001, $\eta_p^2$ = 0.047).

# Discussion

We first found that machine-learning models successfully predicted age when trained on FC, structural MRI, and multimodal datasets. As expected, the structural model predicted age with greater accuracy than the FC model, but the multimodal model outperformed both unimodal models. Second, BAG estimates from all models were significantly elevated in CI participants compared to CN controls. BAG estimates in the FC model were significantly reduced in cognitively normal participants with elevated amyloid, but no structural group differences were observed in presymptomatic stages. Third, interactive relationships were observed, such that greater BAG was associated with greater continuous AD biomarker load in CI, but not in CN, participants. Specifically, in the FC model, such a pattern only appeared in a non-significant interaction predicting CSF pTau/Aβ40. However, in the structural model, these interactions were significantly observed in relation to CSF pTau/Aβ40 and tau PET. In the multimodal model, these same interactions were also observed in addition to a non-significant interaction with amyloid PET. Finally, regarding cognitive relationships, similar interactive patterns were observed, such that in CI participants, greater BAG estimates from structural and multimodal models were associated with lower cognitive performance; however, this relationship was not observed in the FC model.

## Predicting brain age with multiple modalities

We found that a GPR model trained on structural MRI features predicted chronological age in a cognitively normal, amyloid-negative adult sample with an $R^2$ of 0.81. This level of performance is

**Table 2.** Linear regression models predicting functional connectivity (FC)-brain age gap (BAG) (A), S-BAG (B), and FC + S BAG (C). CDR = Clinical Dementia Rating. FD = framewise displacement.

| | A. FC-BAG (df = 348) | | | | B. S-BAG (df = 349) | | | | C. S+FC BAG (df = 348) | | | |
|---|---|---|---|---|---|---|---|---|---|---|---|---|
| | Estimate | SE | p value | $\eta_p^2$ | Estimate | SE | p value | $\eta_p^2$ | Estimate | SE | p value | $\eta_p^2$ |
| Intercept | 30.903 | 3.809 | 0.000 | | 5.830 | 4.899 | 0.235 | | 11.755 | 4.197 | 0.005 | |
| CDR > 0 | 2.169 | 0.997 | 0.030 | 0.013 | 5.105 | 1.287 | 0.000 | 0.043 | 4.305 | 1.099 | 0.000 | 0.042 |
| Amyloid+ | −1.640 | 0.776 | 0.035 | 0.013 | 0.900 | 1.002 | 0.369 | 0.002 | 0.060 | 0.855 | 0.944 | 0.000 |
| Age (y) | −0.586 | 0.044 | 0.000 | 0.335 | −0.151 | 0.057 | 0.008 | 0.020 | −0.201 | 0.049 | 0.000 | 0.047 |
| Sex = female | −1.174 | 0.700 | 0.094 | 0.008 | 1.792 | 0.904 | 0.048 | 0.011 | 0.691 | 0.771 | 0.371 | 0.002 |
| Education (y) | −0.006 | 0.127 | 0.964 | 0.000 | −0.155 | 0.164 | 0.345 | 0.003 | −0.152 | 0.140 | 0.276 | 0.003 |
| Mean FD | 5.528 | 5.467 | 0.313 | 0.003 | NA | NA | NA | NA | 4.893 | 6.024 | 0.417 | 0.002 |

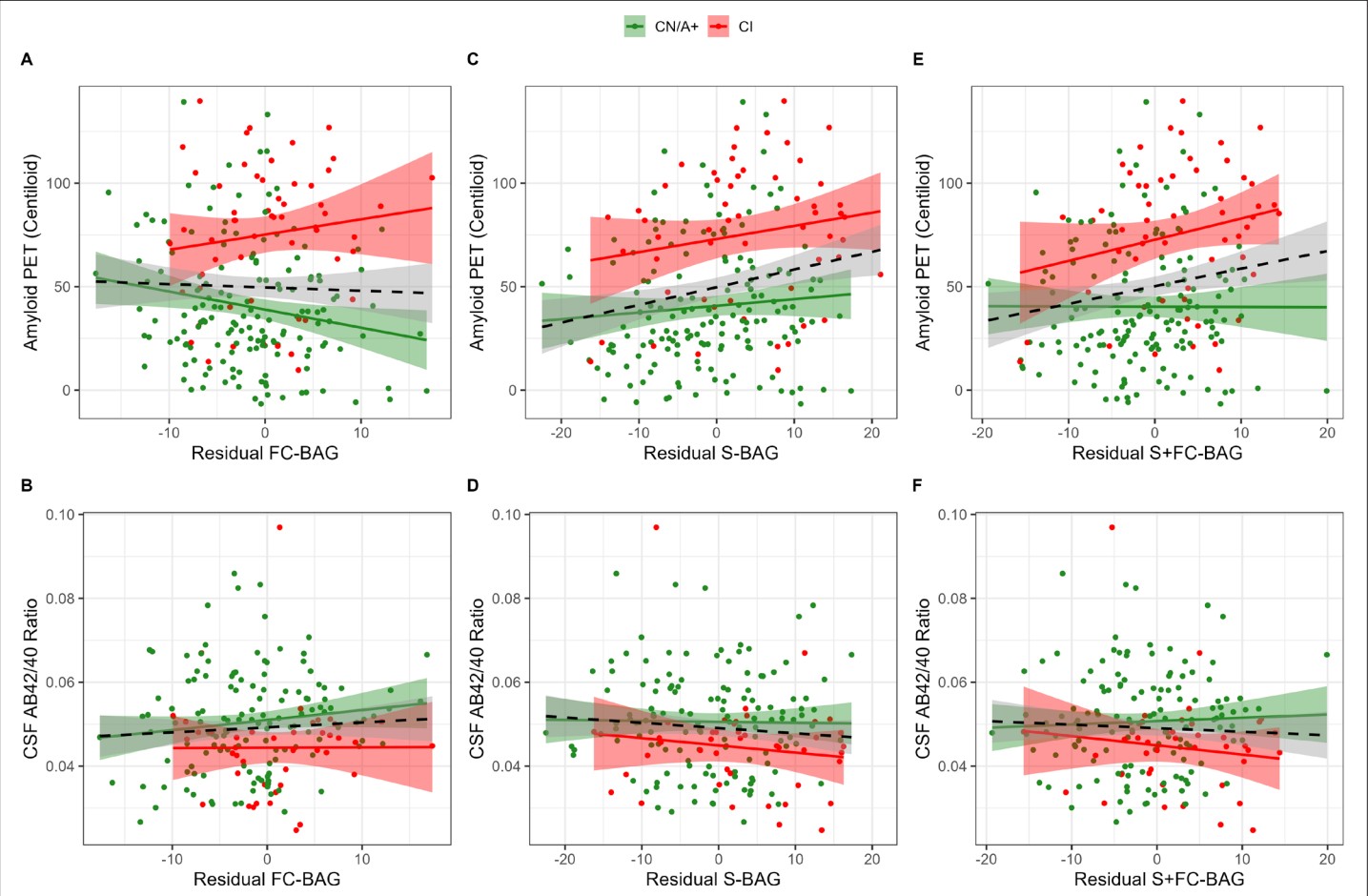

**Figure 3.** Continuous relationships between amyloid biomarkers and functional connectivity (FC-brain age gap [BAG]; **A and B**), structural (S-BAG; **C and D**), and multimodal (S+FC BAG; **E and F**) BAG in the analysis sets. Scatterplots show amyloid PET (**A, C, and E**) and CSF AB42/40 (**B, D, and F**) as a function of residual BAG (controlling for true age) in each group. Colored lines and shaded areas represent group-specific regression lines and 95% confidence regions. Dashed black lines represent main effect regression lines across all groups.

comparable to other structural models, which have reported $R^2$s ranging from 0.80 to 0.95 (*Cole and Franke, 2017b*; *Liem et al., 2017*; *Eavani et al., 2018*; *Wang et al., 2019*; *Bashyam et al., 2020*; *Ly et al., 2020*; *Gong et al., 2021*; *Lee et al., 2022*). As previously reported (*Millar et al., 2022*), the FC-trained model predicted age with an $R^2$ of 0.68, again consistent with previous FC models, which have achieved $R^2$s from 0.53 to 0.80 (*Liem et al., 2017*; *Eavani et al., 2018*; *Gonneaud et al., 2021*). Our observation that structural MRI outperformed FC in age prediction is also consistent with previous direct comparisons between modalities (*Liem et al., 2017*; *Eavani et al., 2018*; *Dunås et al., 2021*).

Importantly, however, there was only a modest positive correlation between FC and structural BAG estimates, after correcting for age-related biases, suggesting that functional and structural MRI capture distinct age-related signals. Indeed, the multimodal model outperformed both unimodal models by integrating these complementary signals. These observations, again, are consistent with other recent reports of multimodal age prediction models (*Liem et al., 2017*; *Eavani et al., 2018*; *Engemann et al., 2020*; *Dunås et al., 2021*). Future models may improve age prediction accuracy by combining data from structural, FC, and/or other neuroimaging modalities, several of which may be available in typical MRI sessions of multiple sequences.

## BAG as a marker of cognitive impairment

Structural BAG was elevated by 5.10 years in CI participants compared to CN controls. This effect is comparable to previous structural age prediction models, demonstrating elevations in AD and MCI samples between 5 and 10 years (*Cole and Franke, 2017b*; *Franke and Gaser, 2019*). As previously

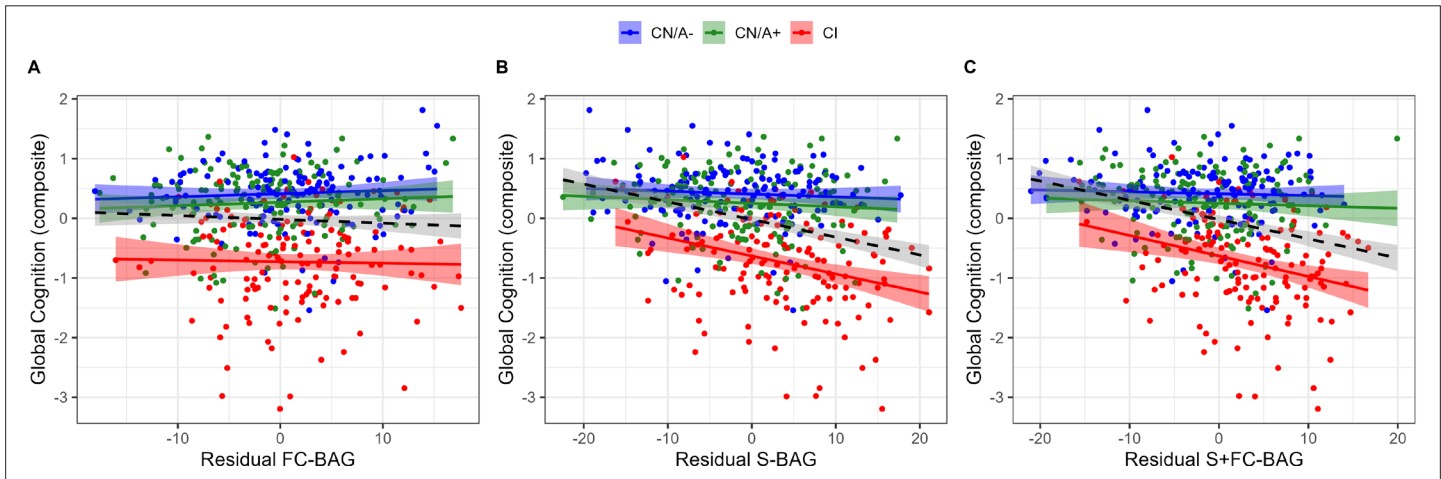

**Figure 4.** Continuous relationships between tau biomarkers and functional connectivity (FC-brain age gap [BAG]; **A and B**), structural (S-BAG; **C and D**), and multimodal (S+FC BAG; **E and F**) BAG in the analysis sets. Scatterplots show Tau PET summary (**A, C, and E**) and log-transformed CSF pTau/Aβ40 (**B, D, and F**) as a function of residual BAG (controlling for true age) in each group. Colored lines and shaded areas represent group-specific regression lines and 95% confidence regions. Dashed black lines represent main effect regression lines across all groups.

**Figure 5.** Continuous relationships between global cognition and functional connectivity (FC-brain age gap [BAG]; **A**), structural (S-BAG; **B**), and multimodal (S+FC BAG; **C**) in the analysis sets. Scatterplots show global cognition as a function of residual BAG (controlling for true age) in each group. Colored lines and shaded areas represent group-specific regression lines and 95% confidence regions. Dashed black lines represent main effect regression lines across all groups.

reported, FC BAG was also elevated in CI participants, but to a relatively smaller extent, i.e., 2.17 years (*Millar et al., 2022*). The multimodal BAG was similarly elevated in CI participants by 5.10 years. Thus, each model is clearly sensitive to group differences in AD status at the symptomatic stage.

Consistent with one previous report (*Lee et al., 2022*), we demonstrated that within the CI participants, BAG estimates were related to individual differences in AD biomarkers and cognitive function. These effects were most pronounced in the structural model, which showed relationships with tau biomarkers and cognition in the CI participants, and the multimodal model, which showed relationships with tau, cognition, and amyloid PET. Thus, age prediction models that include structural MRI (including unimodal and multimodal approaches) may be useful in tracking AD pathological progression and cognitive decline within the symptomatic stage of the disease.

## BAG as a marker of presymptomatic AD

We found that structural and multimodal BAG did not differ between cognitively normal participants with and without amyloid pathology. In cognitively normal participants, structural BAG estimates did not significantly associate with individual differences in any AD biomarkers. Overall, although structural and multimodal BAG estimates track well with some biomarkers of AD pathophysiology, as previously reported (*Lee et al., 2022*), our novel results suggest that these relationships are not observed until the symptomatic stage of the disease, at which point structural changes become more apparent.

As we have previously reported (*Millar et al., 2022*), FC-BAG was lower in presymptomatic AD participants compared to amyloid-negative controls. Extending beyond this group difference, we now also note that FC-BAG was negatively associated with amyloid PET in CN/A+ participants. The combined reduction of FC-BAG in the presymptomatic stage and increase in the symptomatic stage suggest a biphasic functional response to AD progression, which is partially consistent with some prior suggestions (*Jagust and Mormino, 2011*; *Jones et al., 2016*; *Jones et al., 2017*; *Schultz et al., 2017*; *Wales and Leung, 2021*; see *Millar et al., 2022* for a more detailed discussion).

Interpretation of this biphasic pattern is still unclear, although the present results provide at least one novel insight. Specifically, one potential interpretation is that the 'younger' appearing FC pattern in the presymptomatic stage may reflect a compensatory response to early AD pathology (*Cabeza et al., 2018*). This interpretation leads to the prediction that reduced FC-BAG should be associated with better cognitive performance in the preclinical stage. However, this interpretation is not supported by the current results, as FC-BAG did not correlate with cognition in any of the analysis samples.

Alternatively, pathological AD-related FC disruptions may be orthogonal to healthy age-related FC differences, as supported by our previous observation that age and AD are predicted by mostly non-overlapping FC networks (*Millar et al., 2022*). For instance, the 'younger' FC pattern in CN/A+ participants may be driven by hyper-excitability in the preclinical stage (*Harris et al., 2020*; *Ranasinghe et al., 2022*). It is also worth considering that patterns of younger FC-BAG in CN/A+ participants may somehow correspond to a recent observation that patterns of youthful-appearing aerobic glycolysis are relatively preserved in the presymptomatic stage of AD (*Goyal et al., 2022*). Finally, this effect may simply be spuriously driven by poor performance of the FC brain age model, sample-specific noise, and/or statistical artifacts related to regression dilution and its correction (*Butler et al., 2021*). Hence, future studies should attempt to replicate these results in independent samples and further test potential theoretical interpretations.

## BAG as a marker of cognition

Although FC-BAG was not associated with individual differences in a global cognitive composite within any of our analysis samples, greater structural and multimodal BAG estimates were associated with lower cognitive performance within the CI participants. Hence, these estimates may be sensitive markers of cognitive decline in the symptomatic stage. This finding is consistent with previous reports that other structural brain age estimates are associated with cognitive performance in AD (*Eavani et al., 2018*), Down syndrome (*Cole et al., 2017a*), HIV (*Petersen et al., 2021*; *Petersen et al., 2022*), as well as cognitively normal controls (*Richard et al., 2018*).

## Limitations and future directions

The training sets included MRI scans from a range of sites, scanners, and acquisition sequence parameters, which may introduce noise and/or confounding variance into MRI features. We attempted to mitigate this problem by: (1) including only data from Siemens 3T scanners with similar protocols; (2) processing all MRI data through common pipelines and quality assessments; and (3) harmonizing across sites and scanners with ComBat (*Fortin et al., 2017*).

Additionally, the training set (N = 390) was relatively small compared to prior models, which have included training samples over 1,000 (e.g., *Cole et al., 2015*; *Bashyam et al., 2020*). Future studies may further improve model performance by including larger samples of well-characterized participants in the training set.

Although we took appropriate steps to detect and control for AD-related pathology in the CN/A− training sets, we were unable to control for other non-AD pathologies, e.g., Lewy body disease, TDP-43, etc., which may be present.

Structural MRI was quantified using the Desikan atlas (*Desikan et al., 2006*), which, although widely used, provides a relatively coarse parcellation of structural anatomy and, moreover, does not align with the parcellation used to define FC regions (*Seitzman et al., 2020*). Although the structural MRI data still outperformed FC in predicting age, future brain age models may further improve performance by using more refined and harmonized anatomical parcellations to define brain regions.

The sample size of continuous biomarker and cognitive analyses differed across the measures, depending on the availability, and was particularly low for analyses of tau PET. Future studies might improve upon this approach by a larger and more complete biomarker sample.

Moreover, estimates of BAG likely capture variance in early-life factors, which may obscure associations with AD and cognition, especially in cross-sectional designs (*Vidal-Piñeiro et al., 2021*). Future studies may improve the sensitivity of BAG estimates to disease-related markers by testing associations with longitudinal change.

Finally, although the Ances lab controls were relatively diverse, participants in other samples were mostly white and highly educated. Hence, these models may not be generalizable to broader samples. Future models would benefit by using more representative training samples.

## Conclusions

We compared three MRI-based machine-learning models in their ability to predict age, as well as their sensitivity to early-stage AD, AD biomarkers, and cognition. Although FC and structural MRI models were both successful in detecting differences related to healthy aging and cognitive impairment, we note clear evidence that these modalities capture complementary signals. Specifically, FC-BAG was uniquely reduced in cognitively normal participants with elevated amyloid, although the interpretation of this finding still warrants further investigation. In contrast, structural BAG was uniquely associated with biomarkers of AD pathology and cognitive function within the CI participants. Finally, the multimodal age prediction model, which combined FC and structural MRI, further improved the prediction of healthy age differences and also was related to biomarkers and cognition in CI participants. Thus, multimodal brain age models may be useful maximizing sensitivity to AD across the spectrum of disease progression.

## Acknowledgements

We thank the participants for their dedication to this project, Haleem Azmy, Anna Boerwinkle, and Dimitre Tomov for technical and processing support. This manuscript has been reviewed by DIAN Study investigators for scientific content and consistency of data interpretation with previous DIAN Study publications. We acknowledge the altruism of the participants and their families and contributions of the DIAN research and support staff at each of the participating sites for their contributions to this study. We thank the personnel of the Administration, Biomarker, Biostatistics, Clinical, Genetics, and Neuroimaging Cores of the Knight ADRC, as well as the Administration, Biomarker, Biostatistics, Clinical, Cognition, Genetics, and Imaging Cores of DIAN. This research was funded by grants from the National Institutes of Health (P01-AG026276, P01-AG03991, P30-AG066444, 5-R01-AG052550, 5-R01-AG057680, 1-R01-AG067505, 1S10RR022984-01A1) and the BrightFocus Foundation (A2022014F), with generous support from the Paula and Rodger O Riney Fund and the Daniel J Brennan MD Fund. Data collection and sharing for this project was supported by The Dominantly

Inherited Alzheimer Network (DIAN, U19-AG032438) funded by the National Institute on Aging (NIA),the Alzheimer's Association (SG-20–690363-DIAN), the German Center for Neurodegenerative Diseases (DZNE), Raul Carrea Institute for Neurological Research (FLENI), Partial support by the Research and Development Grants for Dementia from Japan Agency for Medical Research and Development, AMED, and the Korea Health Technology R&D Project through the Korea Health Industry Development Institute (KHIDI), Spanish Institute of Health Carlos III (ISCIII), Canadian Institutes of Health Research (CIHR), Canadian Consortium of Neurodegeneration and Aging, Brain Canada Foundation, and Fonds de Recherche du Québec – Santé.

## Additional information

### Group author details

**The Dominantly Inherited Alzheimer Network**
Adam Sarah; Allegri Ricardo; Araki Aki; Barthelemy Nicolas; Bateman Randall; Bechara Jacob; Benzinger Tammie; Berman Sarah; Bodge Courtney; Brandon Susan; Brooks William Bill; Brosch Jared; Buck Jill; Buckles Virginia; Carter Kathleen; Cash Lisa; Chen Charlie; Chhatwal Jasmeer; Mendez Patricio C; Chua Jasmin; Chui Helena; Courtney Laura; Cruchaga Carlos; Day Gregory S; DeLaCruz Chrismary; Denner Darcy; Diffenbacher Anna; Dincer Aylin; Donahue Tamara; Douglas Jane; Duong Duc; Egido Noelia; Esposito Bianca; Fagan Anne; Farlow Marty; Feldman Becca; Fitzpatrick Colleen; Flores Shaney; Fox Nick; Franklin Erin; Joseph-Mathurin Nelly; Fujii Hisako; Gardener Samantha; Ghetti Bernardino; Goate Alison; Goldberg Sarah; Goldman Jill; Gonzalez Alyssa; Gordon Brian; Gräber-Sultan Susanne; Graff-Radford Neill; Graham Morgan; Gray Julia; Gremminger Emily; Grilo Miguel; Groves Alex; Haass Christian; Häsler Lisa; Hassenstab Jason; Hellm Cortaiga; Herries Elizabeth; Hoechst-Swisher Laura; Hofmann Anna; Holtzman David; Hornbeck Russ; Igor Yakushev; Ihara Ryoko; Ikeuchi Takeshi; Ikonomovic Snezana; Ishii Kenji; Jack Clifford; Jerome Gina; Johnson Erik; Jucker Mathias; Karch Celeste; Käser Stephan; Kasuga Kensaku; Keefe Sarah; Klunk William; Koeppe Robert; Koudelis Deb; Kuder-Buletta Elke; Laske Christoph; Levey Allan; Levin Johannes; Li Yan; Lopez MD Oscar; Marsh Jacob; Martins Ralph; Mason Neal S; Masters Colin; Mawuenyega Kwasi; McCullough Austin; McDade Eric; Mejia Arlene; Morenas-Rodriguez Estrella; Morris John; Mountz James; Mummery Cath; Nadkarni Neelesh; Nagamatsu Akemi; Neimeyer Katie; Niimi Yoshiki; Noble James; Norton Joanne; Nuscher Brigitte; Obermüller Ulricke; O'Connor Antoinette; Patira Riddhi; Perrin Richard; Ping Lingyan; Preische Oliver; Renton Alan; Ringman John; Salloway Stephen; Schofield Peter; Senda Michio; Seyfried Nicholas T; Shady Kristine; Shimada Hiroyuki; Sigurdson Wendy; Smith Jennifer; Smith Lori; Snitz Beth; Sohrabi Hamid; Stephens Sochenda; Taddei Kevin; Thompson Sarah; Vöglein Jonathan; Wang Peter; Wang Qing; Weamer Elise; Xiong Chengjie; Xu Jinbin; Xu Xiong

### Competing interests

Tammie LS Benzinger: received doses (AV45, AV1451) and partial support for PET scanning through an investigator-initiated research grant awarded to Washington University from Avid Radiopharmaceuticals (a wholly-owned subsidiary of Eli Lilly and Company). The author received consulting fees from Eisai, Siemens, and received payment for Biogen speaker's bureau. Tammie Benzinger acts as site investigator in clinical trials sponsored by Avid Radiopharmaceuticals, Eli Lilly and Company, Biogen, Eisai, Jaansen and Roche. The author has no other competing interests to declare. Carlos Cruchaga: has received research support from Biogen, EISAI, Alector and Parabon. Carlos Cruchaga is a member of the advisory board of Vivid Genetics, Circular Genomics and Alector. The author has no other competing interests to declare. Anne M Fagan: has received consulting fees from DiamiR and Siemens Healthcare Diagnostics Inc and has received consulting fees for participation on Scientific advisory boards for Roche Diagnostics, Genentech and Diadem. The author has received travel support for in-person attendance at ABC-DS Meeting/Retreat and travel support/honorarium for in-person attendance at Scientific Advisory Board meeting for South Texas Alzheimer's Disease Research Center (ADRC). The author has no other competing interests to declare. Jason J Hassenstab: has received consulting fees from Roche and Parabon Nanolabs. The author has no other competing interests to

declare. Suzanne E Schindler: received personal honoraria for presenting lectures from the University of Wisconsin, St. Luke's Hospital, Houston Methodist Medical Center, personal Honoraria for serving on the Alzheimer Disease Center Clinical Task Force from University of Washington and personal honoraria for serving on the National Centralized Repository for Alzheimer's Disease biospecimen review committee from University of Indiana. The author received travel support from National Institute on Aging grant R01AG070941, and is a board member of the Greater Missouri Alzheimer's Association. The author received plasma Ab42/Ab40 data provided by C2N Diagnostics at no cost. No payments/research funding was provided by C2N Diagnostics. No gifts/financial incentives of any kind have been provided to Dr. Schindler by C2N Diagnostics. The author has no other competing interests to declare. Gregory S Day: received fees for consulting and for acting as Dementia Topic Editor from DynaMed (EBSCO Health) and received fees for consulting, grant writing / implementation Parabon Nanonlabs. The author received payment for CME Content development from PeerView Media and Continuing Education Inc, payment for educational content development and focus group participation from Eli Lilly Co, and payment for continuum manuscript authorship from the American Academy of Neurology. The author received payment for expert testimony in the case of Wernicke encephalopathy from Barrow Law. Gregory S Day acts as Clinical Director for Anti-NMDA Receptor Encephalitis Foundation, Inc. The author has stock holdings at ANI Pharmaceuticals, Inc and stock options at Parabon Nanolabs. The author has no other competing interests to declare. Martin R Farlow: received grants from AbbVie, Eisai, Novartis, ADCS Posiphen, Genentech and Suven Life Sciences (no grant numbers available). The author has received consulting fees from Artery Therapeutics, Avanir, Biogen, Cyclo Therapeutics, Green Valley, Lexeo, McClena, Nervive, Oligomerix, Pinteon, Prothena, Vaxinity, Athira, AZTherapies, Cognition Therapeutics, Gemvax, Ionis, Longeveron, Merck, Neurotrope Biosciences, Otsuka, Proclara and SToP-AD. The author has no other competing interests to declare. The Dominantly Inherited Alzheimer Network: Randall J Bateman: received funding and non-financial support for the DIAN-TU-001 trial from Avid Radiopharmaceuticals, and funding for the DIAN-TU-001 trial from Janssen, Hoffman La-Roche/Genentech, Eli Lilly & Co., Eisai, Biogen, AbbVie and Bristol Meyer Squibb. The author has equity ownership interest in C2N Diagnostics and receives royalty income based on technology (stable isotope labeling kinetics and blood plasma assay) licensed by Washington University to C2N Diagnostics. The author received International Conference Lecture Honoraria from Korean Dementia Association and Conference Lecture Honoraria from Weill Cornell Medical College. The author received support for travel expenses from Alzheimer's Association Roundtable and Duke Margolis Alzheimer's Roundtable. The author participates on an unpaid Advisory Board for Roche Gantenerumab Steering Committee and Biogen - Combination Therapy for Alzheimer's Disease, and participates on an unpaid Scientific Advisory Board for UK Dementia Research Institute at University College London and Stanford University, Next Generation Translational Proteomics for Alzheimer's and Related Dementias. The author receives an income from C2N Diagnostics for serving on the scientific advisory board. The author has received equipment and materials from Avid Radiopharmaceuticals, Eli Lilly & Co, Hoffman La-Roche, Eisai and Janssen. Unrelated to this article, Randall Bateman serves as principal investigator of the DIAN-TU, which is supported by the Alzheimer's Association, GHR Foundation, an anonymous organization and the DIAN-TU Pharma Consortium (Active: Eli Lilly and Company/Avid Radiopharmaceuticals, F. Hoffman-La Roche/Genentech, Biogen, Eisai, and Janssen. Previous: Abbvie, Amgen, AstraZeneca, Forum, Mithridion, Novartis, Pfizer, Sanofi, and United Neuroscience). In addition, in-kind support has been received from CogState and Signant Health. Unrelated to this article Randall Bateman has submitted the US nonprovisional patent application "Methods for Measuring the Metabolism of CNS Derived Biomolecules In Vivo" and provisional patent application "Plasma Based Methods for Detecting CNS Amyloid Deposition". The author has no other competing interests to declare. John C Morris: has received consulting fees from Barcelona Brain Research Center BBRC and Native Alzheimer Disease-Related Resource Center in Minority Aging Research, Ext Adv Board. The author has received payment or honoraria for lectures from Montefiore Grand Rounds, NY and Tetra-Inst ADRC seminar series, Grand Rds, NY. The author has participated on the Research Strategy Council for the Cure Alzheimer's Fund, the Diverse VCID Observational Study Monitoring Board and the LEADS Advisory Board, Indiana University. The author has no other competing interests to declare. The other authors declare that no competing interests exist.

## Funding

| Funder | Grant reference number | Author |
| --- | --- | --- |
| National Institutes of Health | P01-AG026276 | John C Morris |
| National Institutes of Health | P01-AG03991 | John C Morris |
| National Institutes of Health | P30-AG066444 | John C Morris |
| National Institutes of Health | 5-R01-AG052550 | Beau M Ances |
| National Institutes of Health | 5-R01-AG057680 | Beau M Ances |
| National Institutes of Health | U19-AG032438 | Randall J Bateman |
| BrightFocus Foundation | A2022014F | Peter R Millar |
| Alzheimer's Association | SG-20-690363-DIAN | Randall J Bateman |

The funders had no role in study design, data collection and interpretation, or the decision to submit the work for publication.

### Author contributions

Peter R Millar, Conceptualization, Software, Formal analysis, Investigation, Visualization, Methodology, Writing – original draft, Writing – review and editing; Brian A Gordon, Tammie LS Benzinger, Resources, Supervision, Funding acquisition, Project administration, Writing – review and editing; Patrick H Luckett, Conceptualization, Software, Methodology, Writing – review and editing; Carlos Cruchaga, Anne M Fagan, Jason J Hassenstab, Richard J Perrin, Suzanne E Schindler, Ricardo F Allegri, Gregory S Day, Martin R Farlow, Hiroshi Mori, Georg Nübling, Randall J Bateman, John C Morris, Resources, Funding acquisition, Project administration, Writing – review and editing; The Dominantly Inherited Alzheimer Network, Data curation; Beau M Ances, Conceptualization, Resources, Supervision, Funding acquisition, Methodology, Project administration, Writing – review and editing

### Author ORCIDs

Peter R Millar ![ORCID] http://orcid.org/0000-0003-4588-739X
Beau M Ances ![ORCID] http://orcid.org/0000-0003-3862-7397

### Ethics

Human subjects: All participants provided written informed consent in accordance with the Declaration of Helsinki and their local institutional review board. All procedures were approved by the Human Research Protection Office at WUSTL (IRB ID # 201204041).

### Decision letter and Author response

Decision letter https://doi.org/10.7554/eLife.81869.sa1
Author response https://doi.org/10.7554/eLife.81869.sa2

## Additional files

### Supplementary files

• Supplementary file 1. Summary of acquisition parameters for structural T1 and resting-state functional MRI. TR = repetition time, TE = echo time.

• MDAR checklist

### Data availability

This project utilized datasets obtained from the Knight ADRC and DIAN. The Knight ADRC and DIAN encourage and facilitate research by current and new investigators, and thus, the data and code are available to all qualified researchers after appropriate review. Requests for access to the data used in this study may be placed to the Knight ADRC Leadership Committee (https://knightadrc.wustl.

edu/professionals-clinicians/request-center-resources/) and the DIAN Steering Committee (https://dian.wustl.edu/our-research/for-investigators/dian-observational-study-investigator-resources/data-request-form/). Requests for access to the Ances lab data may be placed to the corresponding author. Code used in this study is available at https://github.com/peterrmillar/MultimodalBrainAge (copy archived at swh:1:rev:de233b8fe813f5fcca317ce0a6353047f0dfbb92).

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
