## [Editor Report]

This is a useful study exploring multi-modality brain age (structural plus resting state MRI) in people in the early stages or at risk of Alzheimer's disease. They found solid evidence that people with cognitive impairment had older-appearing brains and that older-appearing brains were related to Alzheimer's risk factors such as amyloid and tau deposition. Their data suggest that the multi-modality brain age model is more accurate than a unimodal structural MRI model.

---

## [Decision Letter]

**Decision letter after peer review:**

Thank you for submitting your article "Multimodal brain age estimates relate to Alzheimer disease biomarkers and cognition in early stages: a cross-sectional observational study" for consideration by *eLife*. Your article has been reviewed by 3 peer reviewers, and the evaluation has been overseen by a Reviewing Editor and Jeannie Chin as the Senior Editor. The following individuals involved in the review of your submission have agreed to reveal their identity: James Cole (Reviewer #1); Didac Vidal-Pineiro (Reviewer #3).

Essential revisions:

The overall consensus from the reviewers is that some of the hypotheses (and conclusions) are supported by the results (specifically, the Vol-BAG model) while others are much weaker (specifically, the FC-BAG models). In consultation between the reviewers, they agreed that the work is "useful" but found the evidence to be "incomplete" for the conclusions as presented.

1. The functional connectivity model provides a poor fit to the data. Given that the FC-based models have been recently published (Millar 2022), the reviewers feel that the authors should temper claims about the importance of the FC-based modelling.

2. The reviewers are skeptical about claims regarding the improvements afforded by multi-modal brain age models. In particular, the bootstrapping analyses actually support the claims that FC data improve the quality of brain age modelling.

3. Overall, the reviewers feel that some of the conclusions, such as the biphasic relationships between functional brain-age models and pathological status, are not strongly supported and need to be tempered. In particular, the reviewers object to referring to results as "marginal".

4. Discussion about the potential implications of sample size would be welcome.

*Reviewer #1 (Recommendations for the authors):*

– In the Methods, they say they used a Gaussian mixture model to define pTau positivity. There are multiple ways to implement GMMs, so more details should be included here.

– The presentation of the MRI Acquisition section in the Methods is not very clear. I suggest the authors consider an alternative format, possibly a supplementary table, where the acquisition details can be more easily appraised. Currently, the acquisition details on the DIAN participants are scarce relative to the ADRC participants.

– Can the authors explain and justify why the fMRI processing included registration to an older-adult template? Could this have caused a bias in the registration accuracy for younger participants?

– It is unclear to me why they chose to perform 10-fold CV and hold-out validation with 1000 bootstraps. To my mind, the latter would have been sufficient. If the authors think including the initial 10-fold CV as well is important, this should be clearly justified.

– It is important that R^2^ is reported for each model performance, not just MAE. As R^2^ is a ratio the values can readily be compared across published studies, while the MAE cannot as it is heavily dependent on the age distribution of the test set. For completeness, they could also consider reporting the Pearson's r correlation between predicted age and age, and the root mean square error as well.

– It is unclear how the model performance comparisons were conducted (Results, pg. 12). While t-tests are mentioned in the text, the exact details should be included in the Methods. My concern here is that the n (sample size) for these comparisons is based on the number of bootstraps (arbitrarily determined by the authors to be 1000), rather than the actual sample size. If that is the case (and Figure 1D suggests it is), this is procedure is incorrect as the sensitivity that these tests have to detect differences would be purely a factor of the number of bootstraps, rather than the number of observations. This means that the experimenter can simply choose to make smaller differences 'significant' simply by adding more bootstraps. This needs to be clarified and corrected if appropriate. One approach to achieve the goal of comparing model performances is to take Pearson's correlations with age from each model and use Z-transformations to test the alternative hypothesis that the correlations are different (e.g., the Steiger test). In that way, the n would be determined by the number of observations, so statistical power would appropriately reflect the data.

– I recommend avoiding saying things like 'marginally lower' when a p-value = 0.110. There's no real evidence that there's a difference here, so hard to say whether it's truly lower or not. Generally avoiding 'trends' at 0.1> p >0.05 is best practice. P-values are important, but effect sizes (with confidence intervals) are often more informative.

– In the Discussion, when comparing age prediction accuracy between studies, it's important not to rely on MAE alone as this can vary greatly as a function of the test set age distribution. They should use R^2^ instead. Where R^2^ is unavailable, it's essential that the age range of each study mentioned in comparison is reported to provide context to the MAE values.

– The evidence for a biphasic relationship between FC-BAG and pre-clinical/clinical status is somewhat over-interpreted, particularly given there was no difference between A+T- and A+T+ people (p=0.11) and the fit of FC brain age is quite poor (i.e., far from the line of identity in Figure 2A). I suggest more caution when discussing this.

– A key limitation that was not mentioned was the small sample size relative to other studies. Perhaps the model performance is similar but given that only MAE is used to compare studies it is hard to draw meaningful conclusions. My impression is that had larger datasets been available, then performance would have improved.

*Reviewer #2 (Recommendations for the authors):*

– As explained in the previous section, the FC-BAG model has very limited prediction power, and therefore the results from the FC-BAG model are not reliable while providing marginal benefit. The FC-BAG results should be moved to the supplementary materials.

– For the FC-BAG models and its relation to other clinical variables, please also another version of the model including mean, median, and maximum head motion during the entire rsfMRI scan as covariates in the model to further ensure the reliability of the results.

– It is not clear to me that the bootstrapped based t-test provides evidence in favor of the Vol+FC-BAG model. In other words, a stacked model combining FC-BAG and Vol-BAG will always perform as well or worse than each model. If the stacking approach takes this into account (not clear in the method section, needs further explanation) the marginal increase in performance can be explained to this unidirectional effect and needs further confirmation based on a model selection step (e.g. using new independent data not used in the training-validation of FC-BAG and Vol-BAG model to compare Vol+FC-BAG and Vol-BAG model).

– After the previous step authors can choose the best performing model (either Vol-BAG or Vol+FC-BAG model) and only present the data for the selected model since results between the two models are redundant and don't add extra information to the reader.

– The analysis of hippocampal volume (specially related to the preclinical AD) needs to be confirmed. To do so, hippocampal volume as well as volumetric features from regions highly correlated with hippocampal volume should be removed from the feature set of Vol-BAG and Vol+FC-BAG models. The models need to be retrained using the same procedure. The relationship between hippocampal volume and the newly calculated Vol-BAG and Vol+FC-BAG values should be reported alongside the current results.

*Reviewer #3 (Recommendations for the authors):*

Find below some recommendations on how (I think) the science in this manuscript might be improved in no particular order.

1. Training sample. It is unclear why one would like to minimize undetected AD pathology (amyloid positivity, that is) in the cognitively healthy training sample as many of these individuals (when Tau negative) have minimal changes in brain structure and function. Since you create a BA "norm" from these individuals, one may benefit from including a bigger, more representative sample using more lenient inclusion criteria. Decisions regarding the training sample can have a big impact on the subsequent interpretation of BA results (e.g. Hwang, 2022, Brain Comm).

2. Group descriptors. It is still a matter of ongoing debate, but I recommend using another descriptor for the amyloid positive group rather than "preclinical AD". Even in the NIAA-AA Research framework from 2018 (Jack Jr.) they only use this tag for individuals that are amyloid and tau positive.

3. Biomarker definition. I am not an expert on biomarkers, but the definition of pTau positivity is uncommon to me "Gaussian mixture model approach to defining pTau positivity based on the CSF pTau/Aβ40 ratio.". Could the authors justify and or cite the correspondent references?

4. Statistical analysis. If I have not misread, the methods section only mentions three test groups (A-, A+, and CDR>0) but the analysis is performed with four groups. This leads to confusion and should be corrected. Also, most higher-level analyses reported in the results are not described in this section. These analyses should be described in the methods section. It is difficult to evaluate whether the performed analyses are appropriate without this description. For example, (lines 323-7) the authors report three different regression models and then a fourth analysis combining the four groups, but only for FC-BAG. This procedure is unclear, not described (as far as I can see), and not justified. Another example is the analysis with NFL which is not mentioned until line 412 (p.20) in the Results section. Also, the authors use different samples for different tests, due to the lack of Biomarker information for some individuals. I suggest adding degrees of freedom/n when reporting the results, so the reader has some information regarding the sample used.

5. The authors are repeating the same analysis in three different modalities (also sometimes they repeat the analyses across several pairs of groups [e.g. lines 323-7]). Thus, I would strongly recommend using some type of multiple comparison corrections.

6. Table 2. The authors should mention what the units in the table represent. Also, I recommend adding df and exact significance values (at least if p >.001).

7. Atlas. The authors used the D-K atlas (not strictly the FS-defined) for BA computation. This is a suboptimal choice, and I would recommend in the future using more fine-grained parcellations. This is not a strong issue, but the choice surprised me since the authors used a 300-ROI parcellation for the rs-fMRI. Also, since the authors use cortical thickness for sampling the cortex, I would not use "Volumetric"-BA as a descriptor.

8. Movement and rs-fMRI. The rs-fMRI preprocessing used might still lead to a signal that is related to movement. Since movement is almost always related to age and disease [and thus can affect both the BA computation and the tests in the test sample], I would suggest taking additional steps in this regard. At the minimum, I would include total motion as an additional covariate in the higher-level analysis and discuss this issue in the limitations section.

9. The results in cognitively healthy samples are largely negative (i.e. do not differ with groups). One possible explanation is that the authors are using cross-sectional samples and thus – even when using BA metrics – have a signal that captures ongoing aging (accelerated aging, if you wish) and baseline (lifelong, preexisting) variability between individuals. The latter may obscure possible existing effects. I recommend the authors acknowledge the limitations of using cross-sectional data to study changes that ought to be longitudinal.

---

## [Author Response]

Essential revisions:The overall consensus from the reviewers is that some of the hypotheses (and conclusions) are supported by the results (specifically, the Vol-BAG model) while others are much weaker (specifically, the FC-BAG models). In consultation between the reviewers, they agreed that the work is "useful" but found the evidence to be "incomplete" for the conclusions as presented.1. The functional connectivity model provides a poor fit to the data. Given that the FC-based models have been recently published (Millar 2022), the reviewers feel that the authors should temper claims about the importance of the FC-based modelling.

Although the reviewers are correct that the FC model indeed provided a relatively poor fit, compared to structural MRI data, a major goal of this project was to test whether each modality (structural MRI and FC) captures unique patterns related to AD progression. As we are primarily motivated to evaluate these models in their associations with AD, it is important to consider that the most accurate BAG models for age prediction are not necessarily the ones that are most sensitive to disease. Indeed, at least one study suggests that models with “moderate” age prediction accuracy might be the most useful in detecting deviation related to disease, as compared to overly “loose” or “tight” age prediction models (Bashyam et al., 2020). We now justify our motivations more clearly in the “Introduction”:

“This project aimed to develop multimodal models of brain-predicted age, incorporating both FC and structural MRI. Participants with presymptomatic AD pathology were excluded from the training set to maximize sensitivity. We hypothesized that BAG estimates would be sensitive to the presence of AD biomarkers and early cognitive impairment. We further considered whether estimates were continuously associated with AD biomarkers of amyloid and tau, as well as cognition. We hypothesized that FC and structural MRI would capture complementary signals related to age and AD. Thus, we systematically compared models trained on unimodal FC, structural MRI, and combined modalities, to test the added utility of multimodal integration in accurately predicting age and whether each modality captures unique relationships with AD biomarkers and cognition.”

Moreover, in the current revision, we aim to focus the discussion on novel associations with this biphasic FC pattern (including the tests of continuous associations with biomarkers and cognition), rather than recapitulating the previously published finding. We also discuss the potential relevance of this result to emerging results from MEG (Ranasinghe et al., 2022) and metabolic PET studies (Goyal et al., 2022). Finally, we now also acknowledge the poor prediction performance of the FC model as a potential spurious explanation of these findings. The discussion of this result now tempers prior claims about the importance of FC:

“As we have previously reported (31), FC-BAG was lower in presymptomatic AD participants compared to amyloid-negative controls. Extending beyond this group difference, we now also note that FC-BAG was negatively associated with amyloid PET in CN/A+ participants. The combined reduction of FC-BAG in the presymptomatic phase and increase in the symptomatic phase suggest a biphasic functional response to AD progression, which is partially consistent with some prior suggestions (77–81) (see ref 31 for a more detailed discussion).

Interpretation of this biphasic pattern is still unclear, although the present results provide at least one novel insight. Specifically, one potential interpretation is that the “younger” appearing FC pattern in the preclinical phase may reflect a compensatory response to early AD pathology (82). This interpretation leads to the prediction that reduced FC-BAG should be associated with better cognitive performance in the preclinical stage. However, this interpretation is not supported by the current results, as FC-BAG did not correlate with cognition in any of the analysis samples.

Alternatively, pathological AD-related FC disruptions may be orthogonal to healthy age-related FC differences, as supported by our previous observation that age and AD are predicted by mostly non-overlapping FC networks (31). For instance, the “younger” FC pattern in CN/A+ participants may be driven by hyper-excitability in the preclinical stage (83,84). It is also worth considering that patterns of younger FC-BAG in CN/A+ participants may somehow correspond to a recent observation that patterns of youthful-appearing aerobic glycolysis are relatively preserved in the preclinical stage of AD (85). Finally, this effect may simply be spuriously driven by poor performance of the FC brain age model, sample-specific noise, and/or statistical artifacts related to regression dilution and its correction (71). Hence, future studies should attempt to replicate these results in independent samples and further test potential theoretical interpretations.”

2. The reviewers are skeptical about claims regarding the improvements afforded by multi-modal brain age models. In particular, the bootstrapping analyses actually support the claims that FC data improve the quality of brain age modelling.

We thank the reviewers for pointing out this flaw in the comparison of model performance. We now test for significant differences between z-transformed Pearson correlations with age in each model using a Williams’s test (as these correlations are dependent in that they share a common variable, age, as opposed to the Steiger test of correlations between different variables). We now report these test results in the

“Comparison of Model Performance” section:

“All models accurately predicted chronological age in the training sets, as assessed using 10-fold cross validation, as well as in the held-out test sets. Overall, prediction accuracy was lowest in the FC model (MAEFC/Train = 8.67 years, R2FC/Train = 0.68, MAEFC/Test = 8.25 years, R2FC/Test = 0.73, see Figure 1A and B). The structural MRI model (MAES/Train = 5.97 years, R2S/Train = 0.81, MAES/Test = 6.26 years, R2S/Test = 0.82, see Figure 1C and D) significantly outperformed the FC model in age prediction accuracy, Williams’s tS vs. FC = 5.39, p <.001. Finally, the multimodal model (MAES+FC/Train = 5.34 years, R2S+FC/Train = 0.86, MAES+FC/Test = 5.25 years, R2S+FC/Test = 0.87, see Figure 1E and F) significantly outperformed both the FC model (Williams’s tS+FC vs. FC = 11.20, p <.001) and the structural MRI model (Williams’s tS+FC vs. S = 5.67, p <.001).”

3. Overall, the reviewers feel that some of the conclusions, such as the biphasic relationships between functional brain-age models and pathological status, are not strongly supported and need to be tempered. In particular, the reviewers object to referring to results as "marginal".

We appreciate the reviewers’ concern over the weak support for some of the results, particularly in the biphasic relationship observed in the multimodal BAG model. In the revised analyses, which focus on a three group comparison (CN/A- vs. CN/A+ vs. CI), the biphasic pattern in the FC-BAG model is clearly reproduced and survives FDR correction for multiple comparisons. However, the previously noted “marginal” biphasic pattern in the S+FC-BAG model is no longer apparent. Thus, we limit our discussion of the biphasic pattern to the FC model, and not the multimodal model. Moreover, we no longer refer to results as “marginal” throughout the revised submission.

4. Discussion about the potential implications of sample size would be welcome.

We agree with the reviewers that the sample size of the training set was relatively small compared to prior models. We now acknowledge this issue as a limitation and an avenue for future development:

“Additionally, the training set (N = 390) was relatively small compared to prior models, which have included training samples over 1000 (e.g., 5,76). Future studies may further improve model performance by including larger samples of well-characterized participants in the training set.”

Reviewer #1 (Recommendations for the authors):– In the Methods, they say they used a Gaussian mixture model to define pTau positivity. There are multiple ways to implement GMMs, so more details should be included here.

We apologize for the lack of clarity in the GMM methods, as multiple reviewers also noted similar concerns. Specifically, we fit a two-component GMM to the continuous pTau data, and then used the model classification to define pTau- and pTau+ participants. However, in order to simplify the analyses and interpretation of results, we have removed the analyses stratifying by pTau positivity and instead focus only on A- vs. A+ participants (see responses below to reviewer #3, comments #3 and 4).

– The presentation of the MRI Acquisition section in the Methods is not very clear. I suggest the authors consider an alternative format, possibly a supplementary table, where the acquisition details can be more easily appraised. Currently, the acquisition details on the DIAN participants are scarce relative to the ADRC participants.

We apologize for the lack of clarity. In the revision, we now provide more specific details on the acquisition parameters for DIAN participants in the main text (“MRI Acquisition” section) and also provide a summary table of the parameters in the Supplementary Material.

“All MRI data were obtained using a Siemens 3T scanner, although there was a variety of specific models within and across studies. As described previously (Millar et al., 2022), participants in the Knight ADRC and Ances lab studies completed one of two comparable structural MRI protocols, varying by scanner (sagittal T1-weighted magnetization-prepared rapid gradient echo sequence [MPRAGE] with repetition time [TR] = 2400 or 2300 ms, echo time [TE] = 3.16 or 2.95 ms, flip angle = 8° or 9°, frames = 176, field of view = sagittal 256x256 or 240x256 mm, 1-mm isotropic or 1x1x1.2 mm voxels; oblique T2-weighted fast spin echo sequence [FSE] with TR = 3200 ms, TE = 455 ms, 256 x 256 acquisition matrix, 1-mm isotropic voxels) and an identical resting-state fMRI protocol (interleaved whole-brain echo planar imaging sequence [EPI] with TR = 2200 ms, TE = 27 ms, flip angle = 90°, field of view = 256 mm, 4-mm isotropic voxels for two 6-minute runs [164 volumes each] of eyes open fixation). DIAN participants completed a similar MPRAGE protocol (TR = 2300 ms, TE = 2.95 ms, flip angle = 9°, field of view = 270 mm, 1.1x1.1x1.2 mm voxels)(McKay et al., 2022). Resting-state EPI sequence parameters for the DIAN participants differed across sites and scanners with the most notable difference being shorter resting-state runs (one 5-minute run of 120 volumes; see Supplementary File 1 for summary of structural and functional MRI parameters) (McKay et al., 2022).”

– Can the authors explain and justify why the fMRI processing included registration to an older-adult template? Could this have caused a bias in the registration accuracy for younger participants?

We apologize for the lack of clarity. In fact, we used two separate templates (one for younger adults and one for older adults). All participants were registered to the age-appropriate template. We now specify this procedure more clearly in “FC Preprocessing and Features”:

“Transformation to an age-appropriate in-house atlas template (based on independent samples of either younger adults or CN older adults) was performed using a composition of affine transforms connecting the functional volumes with the T2-weighted and MPRAGE images.”

– It is unclear to me why they chose to perform 10-fold CV and hold-out validation with 1000 bootstraps. To my mind, the latter would have been sufficient. If the authors think including the initial 10-fold CV as well is important, this should be clearly justified.

We agree with the reviewer that the 10-fold CV and bootstrapped hold-out validation are somewhat redundant. The hold-out validation step was initially performed to facilitate comparison across models. However, several reviewers have critiqued this approach. We have now removed the bootstrapping approach and instead focus on cross validation in the training set, as well as a non-bootstrapped validation in the testing set. We now specify this approach in the “Gaussian Process Regression (GPR)” section:

“Model performance in the training set was assessed using 10-fold cross validation via the Pearson correlation coefficient (r), the proportion of variance explained (R2), the mean absolute error (MAE), and root-mean-square error (RMSE) between true chronological age and the cross-validated age predictions merged across the 10 folds. We then evaluated generalizability of the models to predict age in unseen data by applying the trained models to the held-out test set of healthy controls.”

– It is important that R^2^ is reported for each model performance, not just MAE. As R^2^ is a ratio the values can readily be compared across published studies, while the MAE cannot as it is heavily dependent on the age distribution of the test set. For completeness, they could also consider reporting the Pearson's r correlation between predicted age and age, and the root mean square error as well.

We agree with the reviewer that R^2^ (as well as Pearson’s r and RMSE) are important metrics of model performance, especially for comparison with other studies. We now report these measures in Figure 1, as well as in the “Comparison of Model Performance”:

“All models accurately predicted chronological age in the training sets, as assessed using 10-fold cross validation, as well as in the held-out test sets. Overall, prediction accuracy was lowest in the FC model (MAEFC/Train = 8.67 years, R2FC/Train = 0.68, MAEFC/Test = 8.25 years, R2FC/Test = 0.73, see Figure 1A and B). The structural MRI model (MAES/Train = 5.97 years, R2S/Train = 0.81, MAES/Test = 6.26 years, R2S/Test = 0.82, see Figure 1C and D) significantly outperformed the FC model in age prediction accuracy, Williams’s tS vs. FC = 5.39, p <.001. Finally, the multimodal model (MAES+FC/Train = 5.34 years, R2S+FC/Train = 0.86, MAES+FC/Test = 5.25 years, R2S+FC/Test = 0.87, see Figure 1E and F) significantly outperformed both the FC model (Williams’s tS+FC vs. FC = 11.20, p <.001) and the structural MRI model (Williams’s tS+FC vs. S = 5.67, p <.001).”

– It is unclear how the model performance comparisons were conducted (Results, pg. 12). While t-tests are mentioned in the text, the exact details should be included in the Methods. My concern here is that the n (sample size) for these comparisons is based on the number of bootstraps (arbitrarily determined by the authors to be 1000), rather than the actual sample size. If that is the case (and Figure 1D suggests it is), this is procedure is incorrect as the sensitivity that these tests have to detect differences would be purely a factor of the number of bootstraps, rather than the number of observations. This means that the experimenter can simply choose to make smaller differences 'significant' simply by adding more bootstraps. This needs to be clarified and corrected if appropriate. One approach to achieve the goal of comparing model performances is to take Pearson's correlations with age from each model and use Z-transformations to test the alternative hypothesis that the correlations are different (e.g., the Steiger test). In that way, the n would be determined by the number of observations, so statistical power would appropriately reflect the data.

We thank the reviewer for pointing out this flaw in the comparison of model performance. We now test for significant differences between z-transformed Pearson correlations with age in each model using a Williams’s test (as these correlations are dependent in that they share a common variable, age, as opposed to the Steiger test of correlations between different variables). We now report these test results in the “Comparison of Model Performance” section:

“All models accurately predicted chronological age in the training sets, as assessed using 10-fold cross validation, as well as in the held-out test sets. Overall, prediction accuracy was lowest in the FC model (MAEFC/Train = 8.67 years, R2FC/Train = 0.68, MAEFC/Test = 8.25 years, R2FC/Test = 0.73, see Figure 1A and B). The structural MRI model (MAES/Train = 5.97 years, R2S/Train = 0.81, MAES/Test = 6.26 years, R2S/Test = 0.82, see Figure 1C and D) significantly outperformed the FC model in age prediction accuracy, Williams’s tS vs. FC = 5.39, p <.001. Finally, the multimodal model (MAES+FC/Train = 5.34 years, R2S+FC/Train = 0.86, MAES+FC/Test = 5.25 years, R2S+FC/Test = 0.87, see Figure 1E and F) significantly outperformed both the FC model (Williams’s tS+FC vs. FC = 11.20, p <.001) and the structural MRI model (Williams’s tS+FC vs. S = 5.67, p <.001).”

– I recommend avoiding saying things like 'marginally lower' when a p-value = 0.110. There's no real evidence that there's a difference here, so hard to say whether it's truly lower or not. Generally avoiding 'trends' at 0.1> p >0.05 is best practice. P-values are important, but effect sizes (with confidence intervals) are often more informative.

We appreciate the reviewer’s concern with over-interpretation of non-significant relationships. We now avoid using the terms “marginal” and “trend” throughout the manuscript. We also report effect sizes (partial *η^2^*) for all regression-based analyses.

– In the Discussion, when comparing age prediction accuracy between studies, it's important not to rely on MAE alone as this can vary greatly as a function of the test set age distribution. They should use R^2^ instead. Where R^2^ is unavailable, it's essential that the age range of each study mentioned in comparison is reported to provide context to the MAE values.

We thank the reviewer for pointing out this flaw. We now discuss our model performance in comparison to prior models using R^2^, instead of MAE, in “Predicting Brain Age with Multiple Modalities”:

“We found that a GPR model trained on structural MRI features predicted chronological age in a cognitively normal, amyloid-negative adult sample with an R2 of 0.81. This level of performance is comparable to other structural models, which have reported R2s ranging from 0.80 to 0.95 (Bashyam et al., 2020; Cole and Franke, 2017; Eavani et al., 2018; Gong et al., 2021; Lee et al., 2022; Liem et al., 2017; Ly et al., 2020; Wang et al., 2019). As previously reported (Millar et al., 2022), the FC-trained model predicted age with an R2 of 0.68, again consistent with previous FC models, which have achieved R2s from 0.53 to 0.80 (Eavani et al., 2018; Gonneaud et al., 2021; Liem et al., 2017). Our observation that structural MRI outperformed FC in age prediction is also consistent with previous direct comparisons between modalities (Dunås et al., 2021; Eavani et al., 2018; Liem et al., 2017).”

– The evidence for a biphasic relationship between FC-BAG and pre-clinical/clinical status is somewhat over-interpreted, particularly given there was no difference between A+T- and A+T+ people (p=0.11) and the fit of FC brain age is quite poor (i.e., far from the line of identity in Figure 2A). I suggest more caution when discussing this.

In the revised analyses, which focus on a three group comparison (CN/A- vs. CN/A+ vs. CI), the biphasic pattern in the FC-BAG model is clearly reproduced and survives FDR correction for multiple comparisons. However, the previously noted “marginal” biphasic pattern in the S+FC-BAG model is no longer apparent. Thus, we limit our discussion of the biphasic pattern to the FC model, and not the multimodal model. Moreover, we aim to focus the discussion on novel associations with this biphasic FC pattern (including the tests of continuous associations with biomarkers and cognition), rather than recapitulating the previously published finding. We also discuss the potential relevance of this result to emerging results from MEG (Ranasinghe et al., 2022) and metabolic PET studies (Goyal et al., 2022). Finally, we now also acknowledge the poor prediction performance of the FC model as a potential spurious explanation of these findings. The discussion of this result now reads as follows:

“As we have previously reported (31), FC-BAG was lower in presymptomatic AD participants compared to amyloid-negative controls. Extending beyond this group difference, we now also note that FC-BAG was negatively associated with amyloid PET in CN/A+ participants. The combined reduction of FC-BAG in the presymptomatic phase and increase in the symptomatic phase suggest a biphasic functional response to AD progression, which is partially consistent with some prior suggestions (77–81) (see ref 31 for a more detailed discussion).

Interpretation of this biphasic pattern is still unclear, although the present results provide at least one novel insight. Specifically, one potential interpretation is that the “younger” appearing FC pattern in the preclinical phase may reflect a compensatory response to early AD pathology (82). This interpretation leads to the prediction that reduced FC-BAG should be associated with better cognitive performance in the preclinical stage. However, this interpretation is not supported by the current results, as FC-BAG did not correlate with cognition in any of the analysis samples.

Alternatively, pathological AD-related FC disruptions may be orthogonal to healthy age-related FC differences, as supported by our previous observation that age and AD are predicted by mostly non-overlapping FC networks (31). For instance, the “younger” FC pattern in CN/A+ participants may be driven by hyper-excitability in the preclinical stage (83,84). It is also worth considering that patterns of younger FC-BAG in CN/A+ participants may somehow correspond to a recent observation that patterns of youthful-appearing aerobic glycolysis are relatively preserved in the preclinical stage of AD (85). Finally, this effect may simply be spuriously driven by poor performance of the FC brain age model, sample-specific noise, and/or statistical artifacts related to regression dilution and its correction (71). Hence, future studies should attempt to replicate these results in independent samples and further test potential theoretical interpretations.”

– A key limitation that was not mentioned was the small sample size relative to other studies. Perhaps the model performance is similar but given that only MAE is used to compare studies it is hard to draw meaningful conclusions. My impression is that had larger datasets been available, then performance would have improved.

We agree with the reviewer that the sample size of the training set was relatively small compared to prior models. We now acknowledge this issue as a limitation and an avenue for future development:

“Additionally, the training set (N = 390) was relatively small compared to prior models, which have included training samples over 1000 (e.g., 5,76). Future studies may further improve model performance by including larger samples of well-characterized participants in the training set.”

Reviewer #2 (Recommendations for the authors):– As explained in the previous section, the FC-BAG model has very limited prediction power, and therefore the results from the FC-BAG model are not reliable while providing marginal benefit. The FC-BAG results should be moved to the supplementary materials.

Although FC performed relatively poorly in predicting age, a major goal of this project was to test whether each modality (structural MRI and FC) captures unique patterns related to AD progression. In fact, the FC model indeed captures a unique pattern in that it is reduced in CN/A+ participants, but increased in CI participants, which stands in contrast to patterns observed in the S-BAG model. We view this as an important observation, which belongs in the main text, rather than a supplementary analysis. We now justify our motivations more clearly in the “Introduction”:

“This project aimed to develop multimodal models of brain-predicted age, incorporating both FC and structural MRI. Participants with presymptomatic AD pathology were excluded from the training set to maximize sensitivity. We hypothesized that BAG estimates would be sensitive to the presence of AD biomarkers and early cognitive impairment. We further considered whether estimates were continuously associated with AD biomarkers of amyloid and tau, as well as cognition. We hypothesized that FC and structural MRI would capture complementary signals related to age and AD. Thus, we systematically compared models trained on unimodal FC, structural MRI, and combined modalities, to test the added utility of multimodal integration in accurately predicting age and whether each modality captures unique relationships with AD biomarkers and cognition.”

– For the FC-BAG models and its relation to other clinical variables, please also another version of the model including mean, median, and maximum head motion during the entire rsfMRI scan as covariates in the model to further ensure the reliability of the results.

We agree with the reviewer (as well as Reviewer #3) that appropriate consideration and control for head motion artifact is a critical element in analysis of FC data. Hence, we now include mean framewise displacement (FD) as an additional covariate in all statistical analyses involving the FC and multimodal (S+FC) BAG estimates. We do not include median and maximum, as suggested by the reviewer, in order to minimize potential multi-collinearity in our regression models. As noted in “Statistical Analysis”:

“Given the potential confounding influence of head motion on FC-derived measures (60,76,77), we also included mean FD as an additional covariate of non-interest in the FC and S+FC models.”

– It is not clear to me that the bootstrapped based t-test provides evidence in favor of the Vol+FC-BAG model. In other words, a stacked model combining FC-BAG and Vol-BAG will always perform as well or worse than each model. If the stacking approach takes this into account (not clear in the method section, needs further explanation) the marginal increase in performance can be explained to this unidirectional effect and needs further confirmation based on a model selection step (e.g. using new independent data not used in the training-validation of FC-BAG and Vol-BAG model to compare Vol+FC-BAG and Vol-BAG model).

We appreciate the reviewer’s concern and agree that it is important to demonstrate that increases in model performance are meaningful, rather than driven by unidirectional effects of adding more features and/or capitalizing on chance. Thus, we performed a supplementary analysis, in which we combined the fully trained structural MRI brain age model with a model trained on “reshuffled” FC features using the same stacking approach in 1000 bootstrap samples. Thus, the distribution of R^2^ in this analysis reflects the expected range of model performance from adding unrelated FC features to the structural brain age model. In fact, most of the bootstrapped models performed similarly or worse than the unimodal structural model (see Figure 1—figure supplement 4), suggesting that our stacking approach does not have a unidirectional effect of improvement from adding unrelated features. No simulation achieved performance as high or greater than the fully trained S+FC model, suggesting that the modestly sized increase in the stacked multimodal model (compared to the unimodal structural MRI model) is driven by meaningful age-related FC signal, rather than by simply capitalizing on chance in a larger feature set. We now describe this analysis in

“Comparison of Model Performance” and Figure 1—figure supplement 4:

“It is possible that the modest increase in the multimodal model was due to capitalizing on noise, simply by adding more features to the structural model. Hence, we also compared the observed R2S+FC to a bootstrapped distribution of R2 performance estimates from 1000 resamples using a model in which the original structural MRI model was stacked with a model trained on randomly reshuffled FC features. Thus, this distribution represents the expected improvements in model performance from simply adding new features to the structural MRI model with the stacked approach. The observed R2S+FC outperformed all R2 estimates from this bootstrapped distribution (p < 0.001, see Figure 1—figure supplement 4), suggesting that the modest increase in model performance observed in the stacked multimodal (S+FC) model over the unimodal structural model is due to meaningful age-related FC signal, rather than capitalizing on noise in a larger feature set.”

– After the previous step authors can choose the best performing model (either Vol-BAG or Vol+FC-BAG model) and only present the data for the selected model since results between the two models are redundant and don't add extra information to the reader.

Although our revised and supplementary analyses support the selection of the S+FC BAG model for most accurate prediction of age, as noted above in the response to comment #1, a major goal of this project was to test whether each modality (structural MRI and FC) captures unique patterns related to AD progression. As we are primarily motivated to evaluate these models in their associations with AD, it is important to consider that the most accurate BAG models for age prediction are not necessarily the ones that are most sensitive to disease. In fact, at least one study suggests that models with “moderate” age prediction accuracy might be the most useful in detecting deviation related to disease, as compared to overly “loose” or “tight” age prediction models (Bashyam et al., 2020). We now justify our motivations more clearly in the

“Introduction”:

“This project aimed to develop multimodal models of brain-predicted age, incorporating both FC and structural MRI. Participants with presymptomatic AD pathology were excluded from the training set to maximize sensitivity. We hypothesized that BAG estimates would be sensitive to the presence of AD biomarkers and early cognitive impairment. We further considered whether estimates were continuously associated with AD biomarkers of amyloid, tau, and neurodegeneration (Jack et al., 2016), as well as cognitive function. We hypothesized that FC and structural MRI would capture complementary signals related to age and AD. Thus, we systematically compared models trained on unimodal FC, structural MRI, and combined modalities, to test the added utility of multimodal integration in accurately predicting age and whether each modality captures unique relationships with AD biomarkers and cognition.”

– The analysis of hippocampal volume (specially related to the preclinical AD) needs to be confirmed. To do so, hippocampal volume as well as volumetric features from regions highly correlated with hippocampal volume should be removed from the feature set of Vol-BAG and Vol+FC-BAG models. The models need to be retrained using the same procedure. The relationship between hippocampal volume and the newly calculated Vol-BAG and Vol+FC-BAG values should be reported alongside the current results.

We agree with the reviewer that the associations between hippocampal volume and S-BAG and S+FC-BAG create problems for interpretation, as the S-BAG and S+FC-BAG models include hippocampal volume as input features and are thus circular. Moreover, it is more of interest to this study to test associations with the biomarkers associated with earlier Alzheimer disease stages, including amyloid and tau. Thus, in the interest of simplifying the focus of the study, as well as the interpretation of results, we have decided to remove the analyses of the neurodegeneration markers (including hippocampal volume).

Reviewer #3 (Recommendations for the authors):Find below some recommendations on how (I think) the science in this manuscript might be improved in no particular order.1. Training sample. It is unclear why one would like to minimize undetected AD pathology (amyloid positivity, that is) in the cognitively healthy training sample as many of these individuals (when Tau negative) have minimal changes in brain structure and function. Since you create a BA "norm" from these individuals, one may benefit from including a bigger, more representative sample using more lenient inclusion criteria. Decisions regarding the training sample can have a big impact on the subsequent interpretation of BA results (e.g. Hwang, 2022, Brain Comm).

We agree with the reviewer that the composition of the training sample is critical for interpreting outputs from a brain age model. Indeed, this consideration motivated us to train our model in amyloid-negative participants for both theoretical and empirical reasons. Specifically, although individuals in the earliest preclinical stages of AD (i.e. A+T-N-) likely have minimal detectable structural changes, it is possible that structural changes might be observable in later stage participants (i.e., T+, N+) even if they are cognitively normal. Thus, removing all A+ participants from the training set is a conservative approach to minimize the potential influence of presymptomatic AD pathology in any stage.

Further, although amyloid positivity may lead to minimal structural differences, prior work from our lab (and others) suggests that amyloid may be associated with differences in functional connectivity, and critically that presymptomatic amyloid pathology may confound effects that are otherwise interpreted to reflect “healthy aging” (Brier et al., 2014). Thus, if these participants are included in the training set, the FC model would learn to associate these disease-related FC patterns with normative aging. When applied to an analysis set of amyloid-positive participants, such a model would be less likely to identify deviation in the BAG, as those disease-related differences are incorporated into the model of healthy aging.

This argument was recently tested by Ly and colleagues (2020), who compared two brain age models: one trained on amyloid-negative participants vs. another trained on cognitively normal participants regardless of amyloid status. They found that the amyloid-negative trained model was able to detect differences in brain age between an amyloid-positive and amyloid-negative test sets, but the model that did not exclude amyloid-positive participants was not sensitive to this difference. Although this study was limited in that the amyloid-positive and amyloid-negative samples were drawn from separate, unmatched cohorts, it represents an important proof of concept, upon we aim to expand in this paper.

We have now revised the introduction to make the motivation for this design decision more clear:

“One approach to maximize sensitivity of BAG to presymptomatic AD pathology may be to train brain age models exclusively on amyloid-negative participants. As undetected AD pathology might influence MRI measures, and thus confound effects otherwise attributed to “healthy aging” (Brier, Thomas, Snyder, et al., 2014), including the patterns learned by a traditional brain age model, an alternative model trained on amyloid-negative participants only might be more sensitive to detect presymptomatic AD pathology as deviations in BAG. Indeed, one recent study demonstrated that an amyloid-negative trained brain age model (Ly et al., 2020) is more sensitive to progressive stages of AD than a typical amyloid-insensitive model (Cole et al., 2015). However, this comparison included amyloid-negative and amyloid-positive test samples from two separate cohorts, and thus may be driven by cohort, scanner, and/or site differences. To validate the applicability of the brain-predicted age approach to preclinical AD, it is important to test a model’s sensitivity to amyloid status, as well as continuous relationships with preclinical AD biomarkers, within a single cohort. Another recent comparison demonstrated that both traditional and amyloid-negative trained brain age models were similarly related to molecular AD biomarkers, but that further attempts to “disentangle” AD from brain age by including more advanced AD continuum participants in the training sample significantly reduced relationships between brain age and AD markers (Hwang et al., 2022). Thus, in this study we will apply the amyloid-negative training approach to a multimodal MRI dataset, in order to maximize sensitivity to AD pathology in the presymptomatic stage.”

2. Group descriptors. It is still a matter of ongoing debate, but I recommend using another descriptor for the amyloid positive group rather than "preclinical AD". Even in the NIAA-AA Research framework from 2018 (Jack Jr.) they only use this tag for individuals that are amyloid and tau positive.

We have revised our terminology throughout the manuscript and figures to refer to our groups by clinical assessment and molecular categorization (e.g, CN/A-, CN/A+, CI), rather than staged progression terms (e.g., “preclinical AD”).

3. Biomarker definition. I am not an expert on biomarkers, but the definition of pTau positivity is uncommon to me "Gaussian mixture model approach to defining pTau positivity based on the CSF pTau/Aβ40 ratio.". Could the authors justify and or cite the correspondent references?

To clarify, we fit a two-component GMM to the continuous pTau data, and then used the model classification to define pTau- and pTau+ participants. However, in order to simplify the analyses and interpretation of results, we have removed the analyses stratifying by pTau positivity and instead focus only on A- vs. A+ participants (see response below to comment #4).

4. Statistical analysis. If I have not misread, the methods section only mentions three test groups (A-, A+, and CDR>0) but the analysis is performed with four groups. This leads to confusion and should be corrected. Also, most higher-level analyses reported in the results are not described in this section. These analyses should be described in the methods section. It is difficult to evaluate whether the performed analyses are appropriate without this description. For example, (lines 323-7) the authors report three different regression models and then a fourth analysis combining the four groups, but only for FC-BAG. This procedure is unclear, not described (as far as I can see), and not justified. Another example is the analysis with NFL which is not mentioned until line 412 (p.20) in the Results section. Also, the authors use different samples for different tests, due to the lack of Biomarker information for some individuals. I suggest adding degrees of freedom/n when reporting the results, so the reader has some information regarding the sample used.

We apologize for the lack of clarity in the statistical analysis. In the revision, we have improved the clarity of this section in the following ways:

A. We no longer analyze the data using a four group split (i.e., A-T- vs. A+T- vs. A+T+ vs. CI). Instead, we focus on analyses of three groups (CN/A- vs. CN/A+ vs. CI) consistently throughout the study.

B. We now provide more detail on the higher level analyses, in which we test for group differences between the three analysis sets and test continuous associations with biomarkers and cognitive measures:

“Group differences in each BAG estimate were tested using an omnibus analysis of variance (ANOVA) test with follow-up pairwise t tests on age-residualized BAG estimates, using a false discovery rate (FDR) correction for multiple comparisons. Assumptions of normality were tested by visual inspection of quantile-quantile plots (see Figure 2—figure supplement 1). Assumptions of equality of variance were tested with Levene’s test. Linear regression models tested the effects of cognitive impairment (CDR > 0 vs. CDR 0) and amyloid positivity (A- vs. A+) on BAG estimates from each model, controlling for true age (as noted above) and demographic covariates (sex, years of education, and race). Given the potential confounding influence of head motion on FC-derived measures (59,75,76), we also included mean FD as an additional covariate of non-interest in the FC and S+FC models. We tested continuous relationships with AD biomarkers and cognitive estimates using linear regression models, including the same demographic and motion covariates.”

C. As noted in the response to Reviewer #2, comment #3, analyses of neurodegeneration biomarkers are of less interest to this study, compared to earlier biomarkers of amyloid and tau. Thus, analyses of NfL have been removed from the study.

D. We now report degrees of freedom for our regression analyses of group differences (see Table 2). We also report the number of participants in each group with available measures of each biomarker throughout the results, for example:

“355 participants (144 CN/A-, 154 CN/A+, 57 CI) had an available amyloid PET scan and 300 (120 CN/A-, 137 CN/A+, 43 CI) had an available CSF estimate of Aβ42/40.”

5. The authors are repeating the same analysis in three different modalities (also sometimes they repeat the analyses across several pairs of groups [e.g. lines 323-7]). Thus, I would strongly recommend using some type of multiple comparison corrections.

We agree with the reviewer that appropriate correction for multiple comparisons is necessary for these analyses. We now apply a false discovery rate (FDR) correction to the pairwise t tests, as described in “Statistical Analysis”:

“Group differences in each BAG estimate were tested using an omnibus analysis of variance (ANOVA) test with follow-up pairwise t tests on age-residualized BAG estimates, using a false discovery rate (FDR) correction for multiple comparisons.”

6. Table 2. The authors should mention what the units in the table represent. Also, I recommend adding df and exact significance values (at least if p >.001).

Table 2 presents the β estimates and standard error for the terms in the linear regression models predicting each BAG estimate. We now label this information more explicitly with separated columns. Further, we now provide exact p values for all terms and df for each model.

7. Atlas. The authors used the D-K atlas (not strictly the FS-defined) for BA computation. This is a suboptimal choice, and I would recommend in the future using more fine-grained parcellations. This is not a strong issue, but the choice surprised me since the authors used a 300-ROI parcellation for the rs-fMRI. Also, since the authors use cortical thickness for sampling the cortex, I would not use "Volumetric"-BA as a descriptor.

We agree with the authors that the D-K atlas is a relatively coarse anatomical parcellation. However, as these analyses were based on large, existing datasets that had already been processed and QC’ed with a harmonized pipeline, it would require significant effort to re-parcellate and QC the full dataset. Moreover, despite this coarse parcellation, the structural MRI data still predicts age quite well and outperforms the FC data, which of course uses the finer grained set of ROIs. We now acknowledge the choice of the D-K parcellation as a potential limitation and area of future development in the “Limitations”:

“Structural MRI was quantified using the Desikan atlas (Desikan et al., 2006), which although widely used, provides a relatively coarse parcellation of structural anatomy, and moreover, does not align with the parcellation used to define FC regions (Seitzman et al., 2020). Although the structural MRI data still outperformed FC in predicting age, future brain age models may further improve performance by using more refined and harmonized anatomical parcellations to define brain regions.”

Additionally, we now refer to the “volumetric” brain age model as “structural”, e.g., S-BAG, throughout the manuscript and figures.

8. Movement and rs-fMRI. The rs-fMRI preprocessing used might still lead to a signal that is related to movement. Since movement is almost always related to age and disease [and thus can affect both the BA computation and the tests in the test sample], I would suggest taking additional steps in this regard. At the minimum, I would include total motion as an additional covariate in the higher-level analysis and discuss this issue in the limitations section.

We agree with the reviewer (as well as Reviewer #2) that appropriate consideration and control for head motion artifact is a critical element in analysis of FC data. Hence, we now include mean framewise displacement (FD) as an additional covariate in all statistical analyses involving the FC and multimodal (S+FC) BAG estimates. As noted in “Statistical Analysis”:

“Given the potential confounding influence of head motion on FC-derived measures (60,76,77), we also included mean FD as an additional covariate of non-interest in the FC and S+FC models.”

9. The results in cognitively healthy samples are largely negative (i.e. do not differ with groups). One possible explanation is that the authors are using cross-sectional samples and thus – even when using BA metrics – have a signal that captures ongoing aging (accelerated aging, if you wish) and baseline (lifelong, preexisting) variability between individuals. The latter may obscure possible existing effects. I recommend the authors acknowledge the limitations of using cross-sectional data to study changes that ought to be longitudinal.

We appreciate the reviewer’s suggestion and now discuss this issue as a limitation and area of future development:

“Moreover, estimates of BAG likely capture variance in early-life factors, which may obscure associations with Alzheimer disease and cognition, especially in cross-sectional designs (87). Future studies may improve the sensitivity of BAG estimates to disease-related markers by testing associations with longitudinal change.”